# A comprehensive structural, biochemical and biological profiling of the human NUDIX hydrolase family

Jordi Carreras-Puigvert[1], Marinka Zitnik[2,3], Ann-Sofie Jemth[1], Megan Carter[4], Judith E. Unterlass[1], Björn Hallström[5], Olga Loseva[1], Zhir Karem[1], José Manuel Calderón-Montaño[1], Cecilia Lindskog[6], Per-Henrik Edqvist[6], Damian J. Matuszewski[7], Hammou Ait Blal[5], Ronnie P.A. Berntsson [4], Maria Häggblad[8], Ulf Martens[8], Matthew Studham[9], Bo Lundgren[8], Carolina Wählby[7], Erik L.L. Sonnhammer[9], Emma Lundberg[5], Pål Stenmark[4], Blaz Zupan[2,10] & Thomas Helleday[1]

The NUDIX enzymes are involved in cellular metabolism and homeostasis, as well as mRNA processing. Although highly conserved throughout all organisms, their biological roles and biochemical redundancies remain largely unclear. To address this, we globally resolve their individual properties and inter-relationships. We purify 18 of the human NUDIX proteins and screen 52 substrates, providing a substrate redundancy map. Using crystal structures, we generate sequence alignment analyses revealing four major structural classes. To a certain extent, their substrate preference redundancies correlate with structural classes, thus linking structure and activity relationships. To elucidate interdependence among the NUDIX hydrolases, we pairwise deplete them generating an epistatic interaction map, evaluate cell cycle perturbations upon knockdown in normal and cancer cells, and analyse their protein and mRNA expression in normal and cancer tissues. Using a novel FUSION algorithm, we integrate all data creating a comprehensive NUDIX enzyme profile map, which will prove fundamental to understanding their biological functionality.

[1] Division of Translational Medicine and Chemical Biology, Science for Life Laboratory, Department of Molecular Biochemistry and Biophysics, Karolinska Institutet, Stockholm 171 65, Sweden. [2] Faculty of Computer and Information Science, University of Ljubljana, SI-1000 Ljubljana, Slovenia. [3] Department of Computer Science, Stanford University, Palo Alto, CA 94305, USA. [4] Department of Biochemistry and Biophysics, Stockholm University, 106 91 Stockholm, Sweden. [5] Cell Profiling—Affinity Proteomics, Science for Life Laboratory, KTH—Royal Institute of Technology, Stockholm 17165, Sweden. [6] Department of Immunology, Genetics and Pathology, Science for Life Laboratory, 751 85 Uppsala, Sweden. [7] Centre for Image Analysis and Science for Life Laboratory, Uppsala University, Uppsala 751 05, Sweden. [8] Biochemical and Cellular Screening Facility, Science for Life Laboratory, Department of Biochemistry and Biophysics, Stockholm University, Stockholm 171 65, Sweden. [9] Stockholm Bioinformatics Center, Science for Life Laboratory, Department of Biochemistry and Biophysics, Stockholm University, Box 1031, 171 21 Solna, Sweden. [10] Department of Molecular and Human Genetics, Baylor College of Medicine, Houston, TX 77030, USA. Correspondence and requests for materials should be addressed to J.C.-P. (email: jordi.carreras.puigvert@scilifelab.se) or to T.H. (email: thomas.helleday@scilifelab.se)

The nucleoside diphosphates linked to moiety-X (NUDIX) hydrolases belong to a super family of enzymes conserved throughout all species[1,2], originally called MutT family proteins, as MutT was the founding member. The human MutT homolog MTH1, encoded by the *NUDT1* gene, has antimutagenic properties, as it prevents the incorporation of oxidized deoxynucleoside triphosphates (dNTPs) (e.g., 8-oxodGTP or 2-OH-dATP) into DNA[3,4]. The high diversity in substrate preferences of the NUDIX family members suggests that only a few, or potentially only MTH1, is involved in preventing mutations in DNA[5]. The NUDIX domain contains a NUDIX box (Gx$_5$Ex$_5$[UA] xREx$_2$EExGU), which differs to a certain extent among the family members. As their name suggests, the NUDIX hydrolases are enzymes that carry out hydrolysis reactions, substrates of which range from canonical (d)NTPs, oxidized (d)NTPs, non-nucleoside polyphosphates, and capped mRNAs[6]. The first reference to the NUDIX hydrolases, MutT, dates back to 1954[7] and most of what we know about this enzyme family was discovered through careful biochemical characterization by Bessman and colleagues[1,8] in the 1990s and others more recently, which has been extensively reviewed by McLennan[2,9,10]. Despite decades of research, the biological functions of many NUDIX enzymes remain elusive and several members are completely uncharacterized[11]. An initial hypothesis was that the NUDIX enzymes clean the cell from deleterious metabolites, such as oxidized nucleotides, ensuring proper cell homeostasis[1,12]. Work in model organisms on individual NUDIX members has given some insights, but the key cellular roles of these enzymes, apart from MTH1, are yet to be designated[12–14]. As some NUDIX enzymes are reported to be upregulated following cellular stress[15–18], they may be important for survival of cells under these conditions and are therefore potentially good targets for therapeutic intervention, e.g., killing of cancer cells. Studying the NUDIX hydrolase family of enzymes individually may be hampered by their possible substrate and functional redundancies. To address this, we have undertaken a family-wide approach by building the largest collected set of information presented to date on all human NUDIX enzymes, including biochemical, structural, genetic, and biological properties, and using a novel algorithm, FUSION[19], to interrogate their similarities.

## Results

### Structural and domain analysis of human NUDIX hydrolases.

It is critical to define the relationship between structure and activity, in order to better understand biochemical mechanisms at molecular detail. To determine sequence and structural similarities between the human NUDIX hydrolases, we generated consensus phylogenetic trees using sequences of both full-length (Fig. 1a and Supplementary Fig. 1a) and NUDIX fold domains (Supplementary Fig. 1b, c), and analyzed their available crystal structures (Fig. 1a, b)[20,21]. Multiple sequence alignments were carried out using Clustal Omega[22] followed by Bayesian inference tree generation using MrBayes[23]. Although the alignment and phylogenetic tree of the NUDIX fold domain sequences did have some significant differences compared with the full-length analysis (Fig. 1a and Supplementary Fig. 1b), multiple NUDIX protein structures in complex with relevant substrates have revealed that substrate binding is at times directed from residues outside the NUDIX fold domain[24,25] and, therefore, further analysis was carried out on the full-length sequence alignment and phylogenetic tree. The phylogenetic analysis separated full-length human NUDIX proteins into three general classes and one significant outlier (NUDT22). Phylogenetic assignment accurately grouped NUDIX proteins possessing diphosphoinositol polyphosphate phosphohydrolase (DIPP) activity (NUDT3, NUDT4, NUDT10,

and NUDT11)[26,27], which have almost identical sequences as previously reported[28]. Another distinct group is formed by NUDT7, NUDT8, NUDT16, and NUDT19, also in agreement with previously reported alignments[29]. Although there is no available structure for NUDT7 and NUDT8, as described earlier[29], our analysis also suggests a high grade of sequence similarity between these two NUDIX enzymes given their posterior probability score, which is close to 1, and their percent pairwise identity of 36% (Fig. 1a). The related proteins NUDT12 and NUDT13, both containing the SQPWPFPxS sequence motif common in NADH diphosphatases, were mapped together[30]. Another distinct grouping places NUDT14 and NUDT5 together. The domain exchange responsible for forming the substrate recognition pocket of NUDT5 is not present in the deposited structure of NUDT14, which lacks the N-terminal 39 residues[25]. Although possessing both sequence and structural similarity, MTH1 and NUDT15 have a distinct substrate activity determined by key residues within the substrate binding pocket[21]. NUDT2 and NUDT21 are grouped in the phylogenetic tree and both have demonstrated ability to bind Ap4A[31–34]. As no family-wide structural analysis has been performed previously, we generated superimposed structures of the phylogenetically relevant enzymes (Fig. 1a) and also present the individual human NUDIX enzymes by their available structures and corresponding domains (Fig. 1b, c). Despite the similarities in the NUDIX hydrolase domain (green), including the NUDIX box (blue), there were clear differences in the positions of these domains within the individual proteins. Moreover, three of the NUDIX enzymes (namely NUDT12, NUDT13, and DCP2) contained additional annotated domains compared with the rest of the NUDIX family members.

### Substrate redundancy in the NUDIX hydrolase family.

Key to defining the biological role of the NUDIX hydrolases is to have a comprehensive overview of their respective substrate activities. A substantial amount of work has been devoted to determine the substrates for individual NUDIX hydrolases[3,4,35]. Here we wanted to generate a more comprehensive picture of the substrate specificities of the different human NUDIX enzymes by assessing their activities side-by-side, in a reaction buffer with physiological pH, providing a basis for determining their biological function in cells. We successfully expressed and purified 18 of the 22 human NUDIX proteins from *Escherichia coli* (Supplementary Fig. 2a). Attempts to express NUDT8, NUDT13, NUDT19, and NUDT20 as soluble full-length proteins using several different *E. coli* strains, expression conditions, and tags were unsuccessful. We subsequently set up a high-throughput biochemical screen based on the Malachite Green assay[36] (Supplementary Fig. 2b). Using this setup, at low (5 nM) and high (200 nM) enzyme concentrations, with 25 or 50 μM substrate, we screened 52 putative substrates, including already known ones (e.g., oxidized dNTPs). We confirmed published enzymatic activities of MTH1 and other NUDIX hydrolases, and identified several novel substrates (Fig. 2a and Supplementary Fig. 2b, c). Given the large data set, we summarized the overlap in enzymatic activity by a heat map of all the NUDIX enzymes at the highest concentration, as well as a hierarchical clustering excluding the conditions displaying no activity (Fig. 2a, b). In the cases of overlapping substrate activites, a bar plot is provided, allowing for more accurate comparison (Fig. 2c–e). Some significant novel substrates identified for the human NUDIX enzymes are N2-me-dGTP for MTH1, and Ap4, Ap4dT, Ap4G, and p4G as substrates for NUDT2 (Fig. 2a–c and Supplementary Fig. 2c), which were previously reported to be substrates for NUDT2 orthologs. We found that NUDT12 had activity toward a wide range of substrates, confirming an earlier study performed at a higher pH[30]. As expected, NUDT12 shared

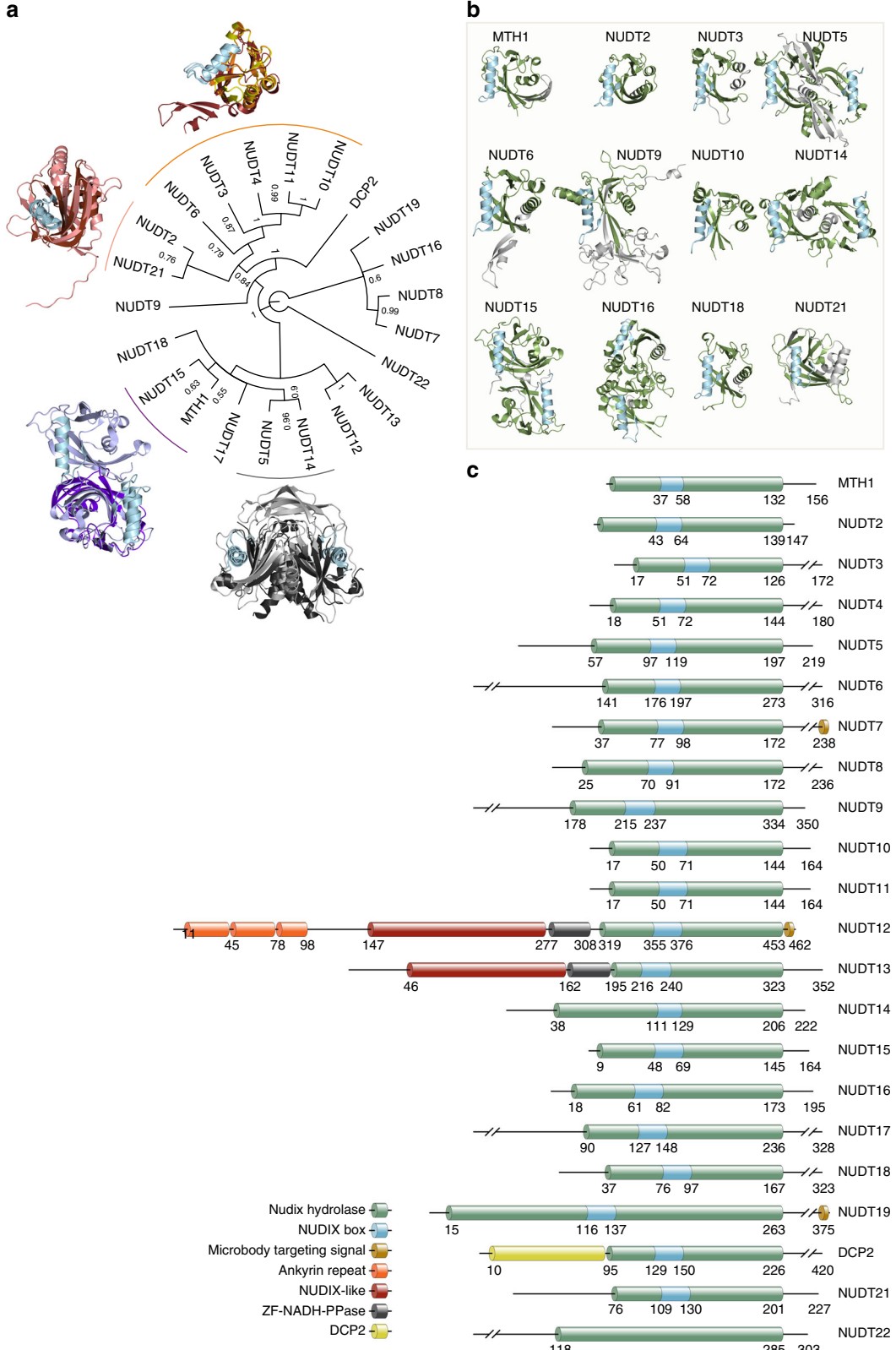

**Fig. 1** Sequence and structural analysis of human NUDIX hydrolases. **a** Consensus phylogenetic tree of full length Human NUDIX proteins with posterior probabilities of each branch provided. Distinct groups with known structures are overlaid for comparison. MTH1 (purple) and NUDT15 (light blue); NUDT5 (gray) and NUDT14 (black); NUDT21 (pink) and NUDT2 (brown); NUDT6 (firebrick red), NUDT3 (yellow), and NUDT10 (orange). **b** Known structures of human NUDIX proteins modeled in cartoon format with the NUDIX box colored in blue, NUDIX fold domain in green, and remaining structure colored in gray. **c** Graphical representation of the different domains within the human NUDIX hydrolases

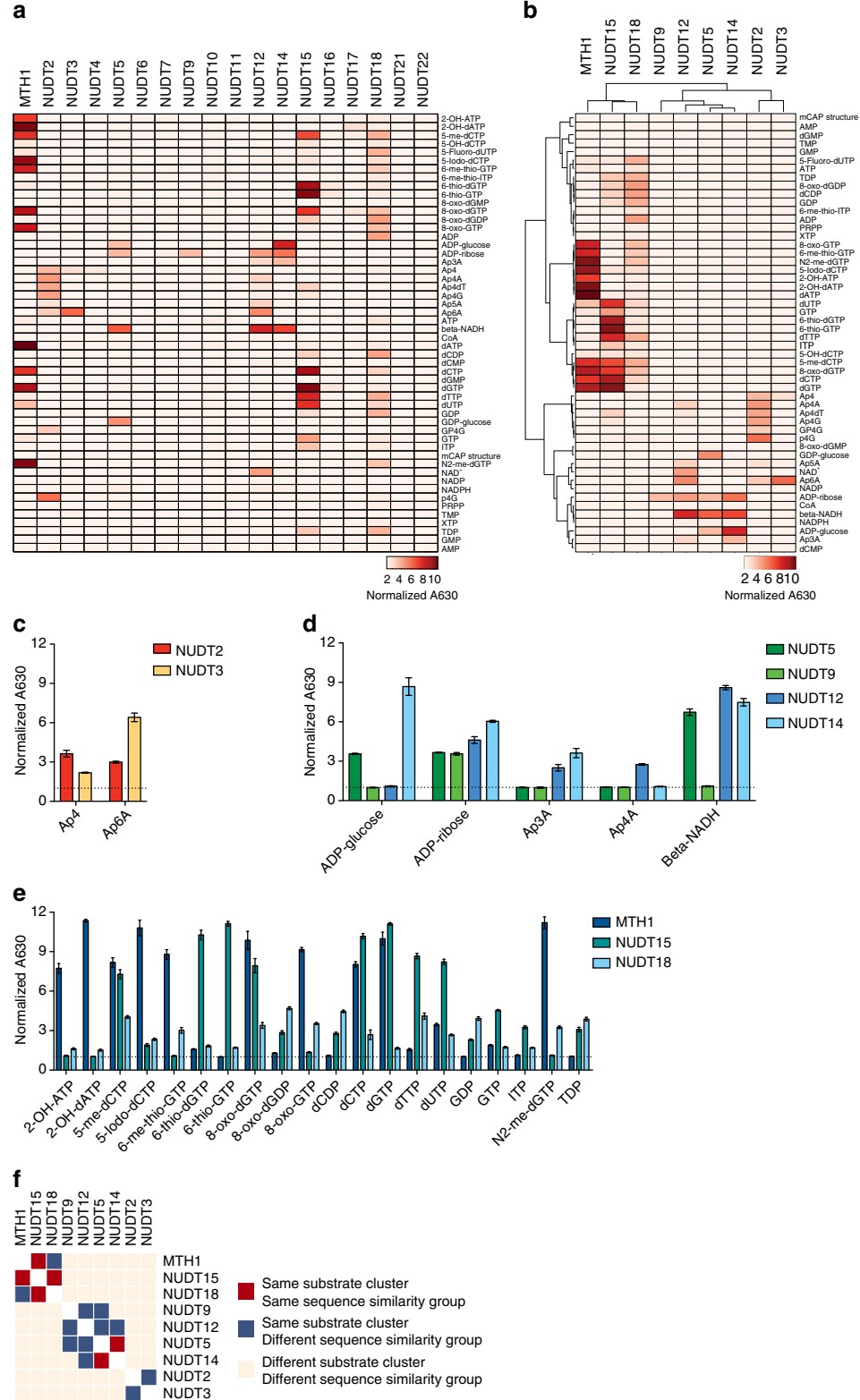

**Fig. 2** Substrate activity of the human NUDIX hydrolases. **a** Activity of 18 human NUDIX hydrolases toward 52 substrates. Activity is represented in a heat map in which the absorbance at 630 nm normalized to untreated controls (this is, without BIP or PPase) is shown. The data represented correspond to the high enzyme concentration condition (200 nM); for the complete data set, see Supplementary Fig. 2d. **b** Hierarchical clustering heat map of the NUDIX hydrolases that displayed activity toward the corresponding substrates. Three distinct clusters appear containing MTH1, NUDT15, and NUDT18; NUDT5, NUDT9, NUDT12, and NUDT14; and NUDT2 and NUDT3. **c** NUDT2 and NUDT3 activity toward their corresponding substrates. **d** NUDT5, NUDT9, NUDT12, and NUDT14 activity toward their corresponding substrates. **e** MTH1, NUDT15, and NUDT18 activity toward their corresponding substrates. **f** Cluster co-assignment matrix comparing sequence similarity grouping and substrate activity clustering

some substrates with NUDT2[30], as well as with NUDT5 and NUDT14. Similar to NUDT5 and NUDT12, NUDT14 showed activity with ADP-glucose and ADP-ribose, in agreement with earlier published results[37], but also with β-NADH and Ap3A, which have not previously been reported (Fig. 2a, b, d and

Supplementary Fig. 2c). NUDT15 showed a rather promiscuous activity over several substrates ranging from modified NTPs including 6-thio-GTP, modified dNTPs such as 5-me-dCTP and 6-thio-dGTP to 8-oxo-dGTP and 8-oxo-dGDP (Fig. 2a, b, e and Supplementary Fig. 2c). Interestingly, our screen failed to identify

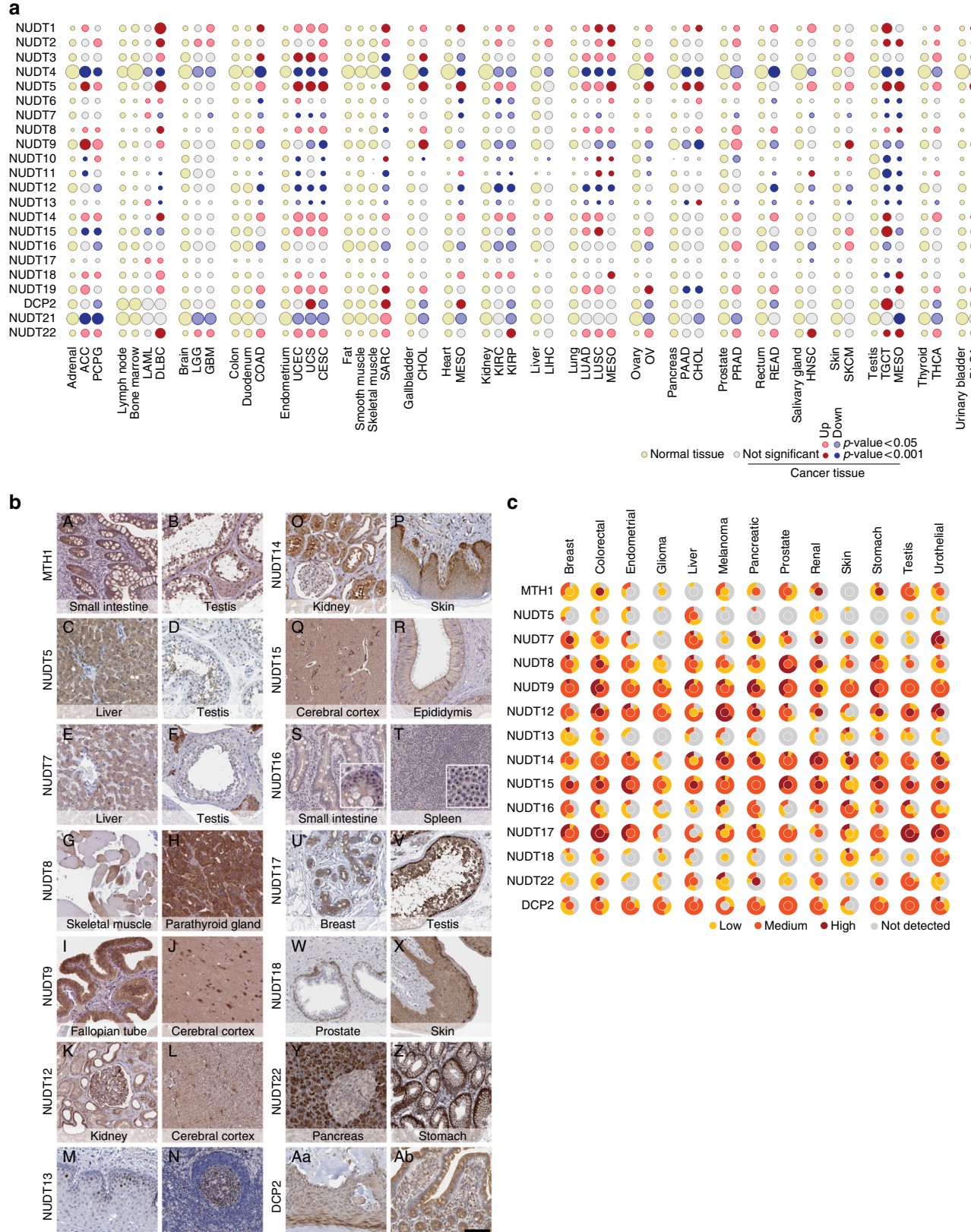

clear substrates for NUDT4, NUDT6, NUDT7, NUDT10, NUDT11, NUDT16, NUDT17, NUDT21, and NUDT22 (Fig. 2a and Supplementary Fig. 2c), indicating that other conditions might be required different than those explored here. NUDT6 is encoded by the fibroblast growth factor antisense RNA and contains the MutT domain; however, as in our case, previous studies have failed to identify a substrate[38,39]. Murine NUDT7 was previously identified as a peroxisomal enzyme with activity toward several Coenzyme A-based substrates[29]. Albeit we used a human purified NUDT7, we cannot explain why we failed to reproduce the reported results. To validate the activity of the DIPP family members, we used their main known substrate[27], 5-PP-InsP5 (Supplementary Fig. 2d), which revealed the expected activity for NUDT3 and NUDT4, but no activity could be detected for NUDT10 and NUDT11.

The hierarchical clustering of the active NUDIX enzymes resembled the one resulting from the sequence analysis (Figs. 1a and 2b), indicating a certain grade of correlation between sequence and substrate activity. To visualize this correlation, we plotted a cluster co-assignment matrix correlation comparing sequence similarity groups and substrate activity clustering (Fig. 2f). The fact that the NUDIX proteins grouped in, either the same sequence similarity group, the same substrate cluster, or both, indicates a correlation between these two features in a subset of members of this enzyme family. However, the phylogenetic tree generated using the NUDIX fold sequences failed to group NUDT2 and NUDT21 (Supplementary Fig. 1b), indicating that the NUDIX fold alignment may not be enough to correctly predict sequence and substrate correlations.

**NUDIX hydrolase gene expression**. Next, we investigated the gene expression of the NUDIX hydrolases in cancer tissues, using the Cancer Genome Atlas (TCGA) and Human Protein Atlas (HPA) databases, and compared cancer vs normal tissues using RNA sequencing data of normal tissues from the HPA[40]. To compare data sets we processed the HPA data according to the TCGA V2 pipeline (see "Expression analysis" in Methods section for reference) and plotted the results using a bubble plot in which the size of the bubble corresponds to the expression levels of each NUDIX gene (Fig. 3a). Up- or downregulation, as well as statistical significance compared with the corresponding normal tissue, is indicated in the figure key. To have a comprehensive overview of normal vs cancer tissues, we paired the available data sets as listed in Supplementary Table 1. In line with previous data, *NUDT1* was significantly overexpressed in almost all of the analyzed cancers[41]. Although *NUDT2* was overexpressed only in

a subset of cancers, *NUDT4* was downregulated in all cancers and appeared to be highly expressed throughout all normal tissues.

Co-expression may reveal an underlying biological function[42]. To determine expression similarities, we used hierarchical clustering to compare the fold-change expression of each tumor type with its corresponding normal tissue (Supplementary Fig. 3a), as well as the expression of each NUDIX enzyme among the normal tissues (Supplementary Fig. 3b). Seemingly, the expression of the NUDIX genes in both normal and cancer samples was tissue dependent, providing a wide spectrum of expression levels (Fig. 3b). However, a distinct cluster appeared when comparing cancer vs normal tissues, which contained *NUDT1*, *NUDT5*, *NUDT8*, *NUDT14*, and *NUDT22* (Supplementary Fig. 3a), confirming a potential role of these NUDIX hydrolases in cancer. Finally, two marked NUDIX genes clusters appeared in normal tissues (Supplementary Fig. 3b).

Our thorough gene expression analysis provides a detailed, but at the same time broad, overview of the NUDIX hydrolases gene expression patterns in healthy as well as cancer tissues, and thereby highlighting important differences across this enzyme family.

**NUDIX hydrolase protein expression**. We determined the diversity of protein expression across organs using immunohistochemistry and tissue microarrays (TMAs), based on manually curated and validated antibodies generated within the HPA pipeline (Fig. 3b, see figure legend for staining details). The protein expression levels are presented as a two-layered circle, where the inner circle represents normal tissues and the color code in the outer circle represents the percentage of cancer tissues that displayed low, medium, high, or not detected expression, allowing for a direct comparison between cancers and their corresponding healthy tissues. MTH1 for instance, appeared to be upregulated in breast cancer and melanoma, whereas downregulated in colorectal cancer, indicating certain divergence between protein and mRNA expression data (Fig. 3a, c). Determining the sub-cellular localization of a protein of interest is important for the understanding of its function. We have used available data from the HPA as well as UniProt to draw a complete overview of the sub-cellular localization of NUDIX hydrolases (Supplementary Fig. 17e). This revealed three main localizations for this family of enzymes: nuclear, mitochondrial and cytosolic, with the exception of NUDT7, NUDT12, and NUDT19, which have known peroxisomal localization.

**Fig. 3** mRNA and protein expression across normal and cancer tissues of the human NUDIX hydrolases. **a** mRNA expression in cancer tissues from the TCGA compared with the non-cancer counterparts from the HPA. Red and blue indicate up- or downregulation, and light brown and gray indicate normal tissue of origin or non-significance in cancer tissue, respectively. A complete list of the cancer types acronyms can be found in the Supplementary Table 3. **b** Immunohistochemical stainings of normal tissues. **a**, **b** MTH1 shows cytoplasmic staining of glandular cells in small intestine and cytoplasmic/nuclear staining seminiferous ducts and testicular Leydig cells. **c**, **d** NUDT5 shows cytoplasmic staining hepatocytes and sperms in testis. **e**, **f** NUDT7 shows cytoplasmic staining of hepatocytes and testicular Leydig cells. **g**, **h** NUDT8 shows patchy cytoplasmic staining of skeletal muscle and parathyroid glandular cells. **i**, **j** NUDT9 shows cytoplasmic staining of glandular cells in the fallopian tube and staining of neurons and neuropil in cortex. **k**, **l** NUDT12 shows cytoplasmic/membranous staining of tubules and glomeruli in kidney and staining of glial cells in cortex. **m**, **n** NUDT13 shows nuclear staining in a subset of squamous epithelial cells in esophagus and in germinal center cells of the lymph node. **o**, **p** NUDT14 shows cytoplasmic and nuclear staining of tubules and glomeruli in kidney and cytoplasmic staining of epidermis (enriched in the basal layer). **q**, **r** NUDT15 shows cytoplasmic/membranous staining of neurons and neuropil in cortex and cytoplasmic/membranous staining of glandular cells in epididymis. **s**, **t** NUDT16 shows nucleolar staining of glandular cells in small intestine and white pulp cells in spleen. **u**, **v** NUDT17 shows cytoplasmic/membranous staining of glandular breast cells and of seminiferous ducts in testis. **w**, **x** NUDT18 shows cytoplasmic and nuclear staining of basal cells of the prostate and in epidermis. **y**, **z** NUDT22 shows cytoplasmic staining of exocrine (strong) and endocrine (weak) pancreatic cells, and cytoplasmic/membranous staining of glandular cells of the stomach. **Aa**, **Ab** DCP2 shows cytoplasmic staining in epidermis, and in stromal and glandular cells of the small intestine. **c** Qualitative assessment graphical representation of the human NUDIX protein expression. The inner circles represent the expression in the normal tissue corresponding to its cancer counterpart. The outer circle represents the percentage of cancers that displayed either not detectable, low, medium, or high protein expression

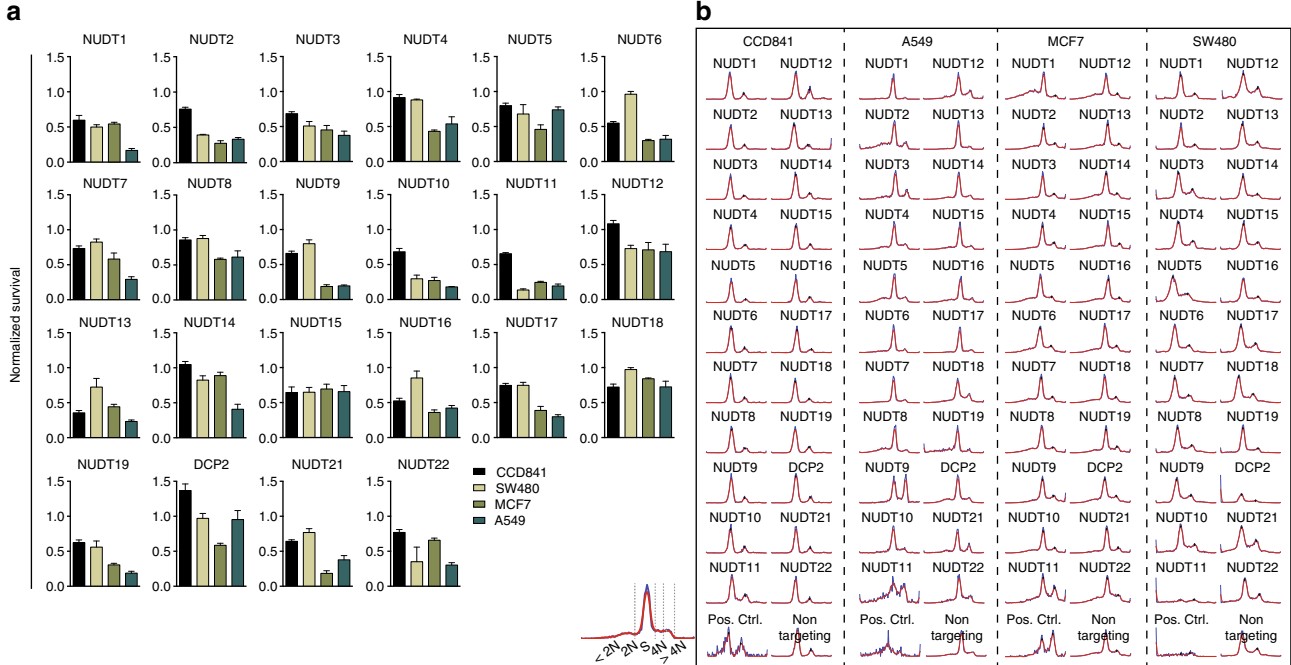

**Fig. 4** Cell viability and cell cycle profiles upon single NUDIX depletion. **a** Survival of CCD841, A549, MCF7, and SW480 cells upon single depletion of the NUDIX enzymes using a pool of four siRNA sequences. The survival was measured by resazurin and normalised to the non-targeting siRNA control. **b** Cell cycle profiles upon single NUDIX knockdown in CCD841, A549, MCF7, and SW480 cells. The histograms were obtained by measuring the integrated intensity of the DNA counterstained with Hoechst and the signal was then processed using PopulationProfiler as described in[46]

**NUDIX hydrolases required for cell survival and cell cycle**. The biological role of the majority of the NUDIX enzymes remains unclear; however, some are implicated in cancer or modulate the response to certain anticancer therapies such as 6-thioguanine[41,43–45]. In order to connect biochemical and biological functions, we small interfering RNA (siRNA)-depleted all human NUDIX proteins and evaluated cell viability and cell cycle distribution (Fig. 4a, b). We used a small panel of cell lines representing three different types of cancers—A549 for lung, MCF7 for breast, and SW480 for colon cancers—as well as the colon epithelial-derived non-cancer cell line CCD841, in which we ran two independent siRNA experiments. As indicated by the high correlation between the knockdown experiments, we achieved a good reproducibility in all four cell lines and, in addition, we obtained a high level of mRNA depletion of each NUDIX, tested in A549 cells by quantitative PCR (qPCR), indicating high confidence results (Supplementary Fig. 4a, b). NUDT1 and NUDT2 depletion, as expected[41,43,44], reduced the proliferation of A549 and MCF7 cells considerably. Interestingly, we identified NUDT10 and NUDT11 to be essential in all three cancer cell lines (Fig. 4a). Of note, given the high sequence similarity between NUDT10 and NUDT11, we acknowledge that the specificity of their corresponding siRNA is not as high as desired. Nonetheless, both knockdowns resulted in a similar lethal phenotype (Fig. 4a). Compared with all other NUDIX enzymes, NUDT13 was essential in CCD841 cells. We analyzed the cell cycle profiles using a DNA content approach[46]. In contrast to the CCD841, the cancer cell lines displayed a wide range of cell cycle effects upon depletion of the different NUDIX enzymes, namely increases in sub-$G_0$/$G_1$ (indicating increase in cell death), arrest in G1 (2 N) or accumulation in $G_2$/M (4 N). We confirm previously known cell cycle perturbations upon NUDIX depletion such as NUDT2 and NUDT5 in cancer cells[43,47,48], characterized by an accumulation in G1 (2 N) phase. These data highlight the potential role of NUDIX hydrolases in cell cycle regulation, either

in a direct manner or through a secondary regulation due to nucleotide pool imbalance, which can lead to replication-slowing DNA lesions[49,50].

**NUDIX genetic interactions uncover biological redundancies**. As some of the NUDIX hydrolases have overlapping biochemical functions, there is also a high likelihood that different proteins within this family are redundant. However, biochemical redundancy may not necessarily equal to a biological redundancy between proteins, as the activity may be distinct under certain biological conditions, or be located to different subcellular compartments. A widely used approach to address this question is the use of functional genomics together with inferred genetic interaction networks[51]. To explore this potential network, we investigated viability and cell cycle perturbations after double siRNA-mediated knockdowns of all the human NUDIX hydrolases in a pairwise manner, thereby producing 276 combinations, in the cell lines CCD841, A549, MCF7, and SW480 (Supplementary Figs. 5 and 7–11). We determined whether the depletion of two NUDIX enzymes had an aggravating, nonsignificant, or alleviating effect on cell viability by normalizing to the corresponding single knockdown controls. Among the several mathematically distinct definitions of genetic interactions or epistasis, many studies[52] provide multiple lines of evidence favoring the multiplicative model; therefore, we decided to use this model in our study. This approach predicts double knockdown viability to be the product of the corresponding single knockdown viability values, i.e., $E(W_{ab}) = W_a W_b$, where a gene pair $(a,b)$, refers to the viability of the two single NUDIX knockdowns and the double knockdown as $W_a$, $W_b$, and $W_{ab}$, respectively. An epistasis interaction score under this definition is then determined as $\epsilon = W_{ab} - E(W_{ab})$ (Fig. 5a). A negative epistasis score suggests an aggravating genetic interaction between two genes, indicating that they likely belong to different pathways, whereas a positive epistasis score is

indicative of alleviating genetic interaction between genes likely to be in the same pathway. Clearly, some of the NUDIX enzymes are epistatic with each other (Fig. 5a and Supplementary Fig. 5b).

To visualize the genetic interactions, we represented them in a network, distinguishing between alleviating (blue) and aggravating (red) genetic interactions (Fig. 5b). We compared the overlap among genetic interaction networks of different cancer cell lines using a stringent 0.05 $\alpha$-cutoff value (Fig. 5c). The resulting Venn

diagrams showed a low overlap of significant genetic interactions among the cancer cell lines, indicating that most of the significant interactions were cell line specific. There was an overlap of four significant interactions between the cancer cell lines and the non-cancerous CCD841 (Fig. 5c), overall indicating weak conservation of both strongly positive and negative genetic interactions among the different cell lines. However, despite the small overlap, we calculated the Spearman's rank correlation of the epistasis scores

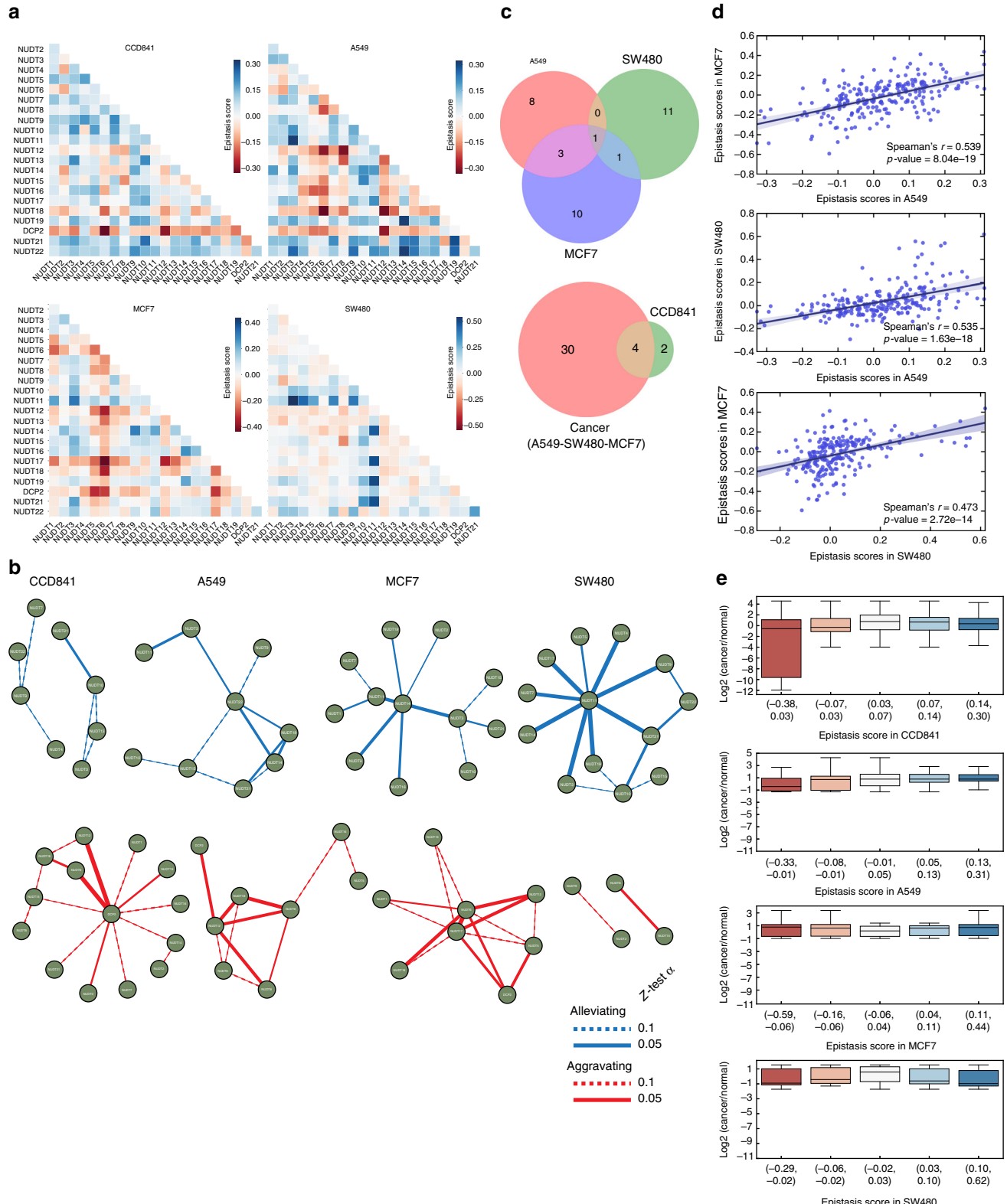

between paired cancer cell lines (Fig. 5d). The positive Spearman's rank score indicated a certain epistasis correlation among the cancer cell lines, namely the knockdown of the same pair of NUDIX enzymes had a similar effect in two different cell lines.

In order to understand the correlation between epistatic interactions and mRNA expression of the NUDIX enzymes in cancer tissues, we compared these two parameters in a box plot (Fig. 5e). We divided the epistasis scores in five bins containing pairs of NUDIX genes. Subsequently, we compared these scores with the log2 mRNA expression of these NUDIX genes in cancer and normal tissues. The NUDIX genes with strongly negative epistatic interactions in CCD841 cells tend to substantially decrease their mRNA expression in cancer tissues. On the contrary, the expression of NUDIX genes with strongly positive epistatic interactions, remained unchanged. As for the cancer cell lines, we compared their epistasis scores to specific cancer tissues resembling their tissue of origin, that is: A549 to LUAD and LUSC, MCF7 to OV and PRAD, and SW480 to COAD.

We next wanted to investigate the correlation between epistatic interactions and sequence similarity, as well as similarity in substrate activity (Supplementary Fig. 6). For each cell line we used box plots to compare full-length and NUDIX fold sequence Patristic distances from our phylogenetic trees, with their epistatic interactions. Lastly, we compared the NUDIX enzymatic activity similarity calculated by Spearman's rank correlation with the epistatic interactions. When comparing full-length sequence distance, for all cell lines, the NUDIX proteins with strong negative interactions also tend to have a lower Patristic distance, which indicates higher sequence similarity (Supplementary Fig. 6a). This was not as clear when comparing NUDIX fold sequence distances (Supplementary Fig. 6b). As for substrate activity similarity compared with epistatic interactions, NUDIX enzymes with negative or aggravating genetic interactions had the highest Spearman's correlation score, mainly in CCD841, but also in A549 and MCF7, but less pronounced in SW480 cells (Supplementary Fig. 6b). A list of NUDIX pairs for each epistasis score bin can be found in Supplementary Data 1.

In addition, we calculated the epistasis scores of the pairwise siRNA-depleted cells depending on their cell cycle distribution (A549 cells in Fig. 6a and rest of cell lines in Supplementary Figs. 7 to 11). We represented each cell cycle phase in one circular network showing interactions with Z-test scores corresponding to a p-value <0.1 (dotted line) and a p-value <0.05 (solid line). We maintained the position of the NUDIX enzymes fixed for better visual assessment of the differences in genetic interactions. This time, instead of classifying the interactions into alleviating or aggravating, we interpreted the cell cycle interactions as percentage of cells increasing (blue) or decreasing (brown) in a given cell cycle phase. For example, in A549 cells, as it is represented by a solid blue edge between the NUDT5 and

NUDT8 nodes, as well as NUDT5 and DCP2 nodes, the double knockdown resulted in an increased number of cells in sub-$G_0/G_1$ phase, indicating increased cell killing (Fig. 6b, c), which is in concordance with decreased survival (Supplementary Fig. 5b). On the other hand, double knockdown of NUDT1 and NUDT12, resulted in a decreased number of cells in G1 phase, especially compared with the single NUDT1 knockdown (Fig. 6b, d). We generated graphical representations of the cell cycle profiles, presented by histograms of cell counts versus DNA content and therefore cell cycle phase (Supplementary Figs. 7–11). In addition, we provide heat maps representing the amount of cells in each cell cycle phase for each single and double knockdowns (Supplementary Fig. 13 and Supplementary Data 2). Similar to the survival epistasis, in which there was a slight overlap among the cancer and CCD841 cells, we also observed some overlap among the genetic interactions (network edges) in each cell cycle phase (Supplementary Fig. 12b). Altogether, the genetic interaction networks extracted from the biological data clearly demonstrate that there is a certain redundancy within the NUDIX family, not only related to cell survival, but also in regulating the cell cycle.

**Réd inferred NUDIX networks reveal potential directionality.** Next, by analysing functional dependencies between the NUDIX genes, we wanted to know whether quantitative genetic interaction measurements could be used to provide detailed information regarding the structure of the underlying biological pathways. For this, we made use of the analytical tool Réd[53], that uses phenotypic measurements of single and double knockdowns to automatically reconstruct detailed pathway structures. We applied Réd to our cell viability data set and used it to calculate relationships between NUDIX genes based on epistasis (Fig. 7). Réd searches for networks that encode independence assumptions supported by genetic interaction measurements. For example, if a given NUDIX gene A appears fully epistatic to a NUDIX gene B, the network should indicate that the cell viability is independent of the activity level of B given the activity level of A, an independence property that is encoded by a linear pathway structure.

We conducted a series of computational experiments to estimate which relationships hold between the NUDIX genes in the different cancer cell lines and in non-cancer cells (Fig. 7 and Supplementary Fig. 14). We systematically evaluated genetic interactions among all combinations of NUDIX genes and used the precise cell viability measurements to distinguish between epistasis and full or partial dependence between two genes[54]. Réd provided probabilistic estimates for each of the four possible network structures on two genes, which we studied independently for each cell line (Fig. 7a and Supplementary Fig. 14a–c). We then tested how the map of the NUDIX family wiring diagram breaks

---

**Fig. 5** Survival genetic interactions between NUDIX genes. **a** Genetic interactions between NUDIX genes in the four cell lines, CCD841, and cancer cell lines A549, MCF7, and SW480. A genetic interaction was assigned to pairs of genes based on deviation of cell viability of the double knockdown from cell viability of the double knockdown that would be expected if the genes were not interacting. The expected viability was determined with a multiplicative null function. The interaction maps include negative (or aggravating) interactions, as well as positive (or alleviating interactions). Alleviating interactions, shown in blue, suggest that certain NUDIX product operate in concert or in series within the same pathway. **b** Statistically significant genetic interactions between NUDIX genes in the four cell lines, CCD841, and cancer cell lines A549, MCF7, and SW480 are visualized using networks. For each gene pair, the genetic interaction was assessed by using a two-tailed Z-test $\alpha = 0.1$ (dotted line and solid line) or $\alpha = 0.05$ (solid line only). Shown are genetic interactions whose values are significantly larger (indicating alleviating interaction) or significantly smaller (indicating aggravating interaction) than values in the 90% (dotted line and solid line), or 95% (solid line only) of interaction density in the respective cell line. **c** The overlap of significant genetic interactions from **b** ($\alpha = 0.05$) is shown using Venn diagrams. The size of each circle in the diagram is proportional to the number of significant genetic interactions in the respective cell line. **d** Scatter plot indicating the correlation between each epistasis scores corresponding to each cell line, Spearman's correlation indicates high similarity. **e** Box plots comparing log2 mRNA expression in cancer vs normal tissues, and epistasis score. Five epistasis score bins were used to classify the NUDIX genetic interactions. The list of each NUDIX interaction can be found in Supplementary Data 1

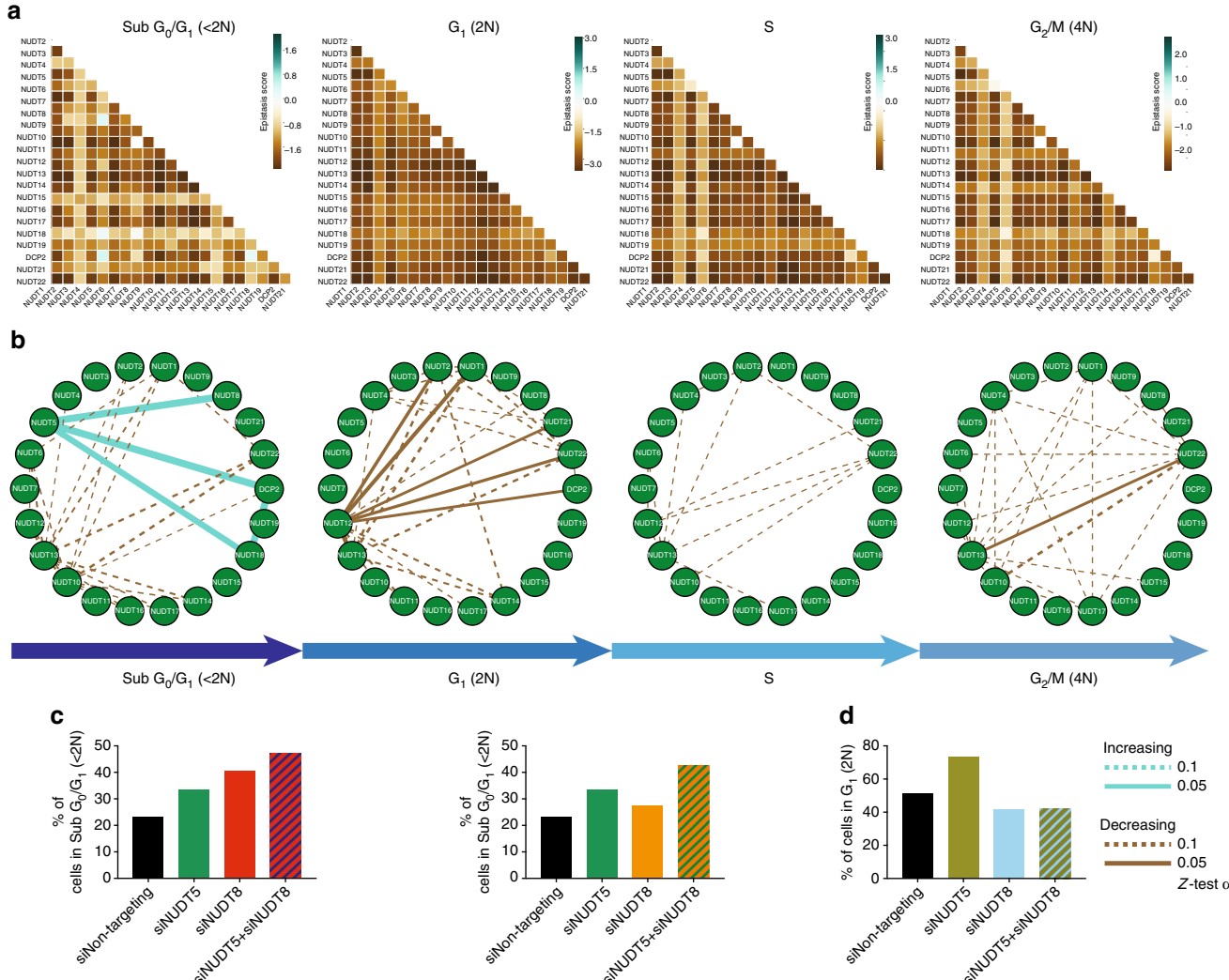

**Fig. 6** Cell cycle genetic interactions between NUDIX genes. **a** Cell cycle-based interactions between NUDIX genes in the A549 cell line. The interaction maps visualize interactions determined based on the fraction of pairwise siRNA-depleted cells in each cell cycle phase. Shown is one interaction map per cell cycle phase. In each map, an interaction score was assigned to a pair of genes based on the difference between the observed cell fraction of the double knockdown and the expected cell fraction of the double knockdown. The expected cell fraction was determined using a multiplicative null model estimating the cell fraction of a double knockdown that would be expected if the genes were not interacting. The interaction maps include negative (or aggravating) interactions in brown, as well as positive (or alleviating) interactions in green. Alleviating interactions suggest that certain NUDIX product operate in concert or in series within the same pathway. **b** Statistically significant cell-cycle-based interactions between NUDIX genes in the A549 cell line are visualized using circular networks. The panel shows one network for each cell cycle phase. For each gene pair, the interaction was assessed by using a two-tailed $Z$-test ($\alpha = 0.1$). Edges in each network represent interactions whose values are significantly larger (indicating alleviating interaction) in cyan or significantly smaller (indicating aggravating interaction) in brown, than values in the 90% of interaction probability density. The interactions were selected independently and separately for each cell cycle phase in the A549 cell line. The width of network edges stands for statistical significance. **c** Bar charts indicating the increase in % of cells in $SubG_0/G_1$ (<2 N) phase when NUDT5 and NUDT8, as well as NUDT5 and DCP2 are co-depleted. **d** Bar chart indicating the decrease in % of cells in $G_1$ (2 N) phase when NUDT1 and NUDT12 are co-depleted. The % of cells in each cell cycle phase were obtained by measuring the integrated intensity of the DNA counterstained with Hoechst, the signal was then processed using PopulationProfiler, as previously described[46]

down in the context of a particular cancer cell line. To provide a comprehensive view of pairwise NUDIX relationships in cancer cells that diverge from those identified in non-cancer cells, we visualized them in differential color maps (Fig. 7b and Supplementary Fig. 14d, e). An alternative complementary view is to examine relationships that are common to all three considered cancers. Many relationships indicating independent downstream effects on the phenotype appeared to remain conserved when comparing interaction maps from A549, SW480, and MCF7, which differ from the ones we found in CCD841 (Supplementary Fig. 14f, g).

To model epistasis at the level of the entire NUDIX family, we used Réd to infer an interaction network in non-cancer cells (Fig. 7c) and, in addition, using common inference data from A549, SW480, and MCF7 cells, Réd predicted the NUDIX cancer epistasis network (Fig. 7d) with both networks clearly in contrast to each other. To assess the stability of the edges in the inferred networks, we tested them against small perturbations of the input data (Supplementary Fig. 15). We used solid lines to visualize confident edges, which were robust to small data perturbations and exhibited low sensitivity to variations of prediction model parameters. We used dashed lines to show edges, which exhibited

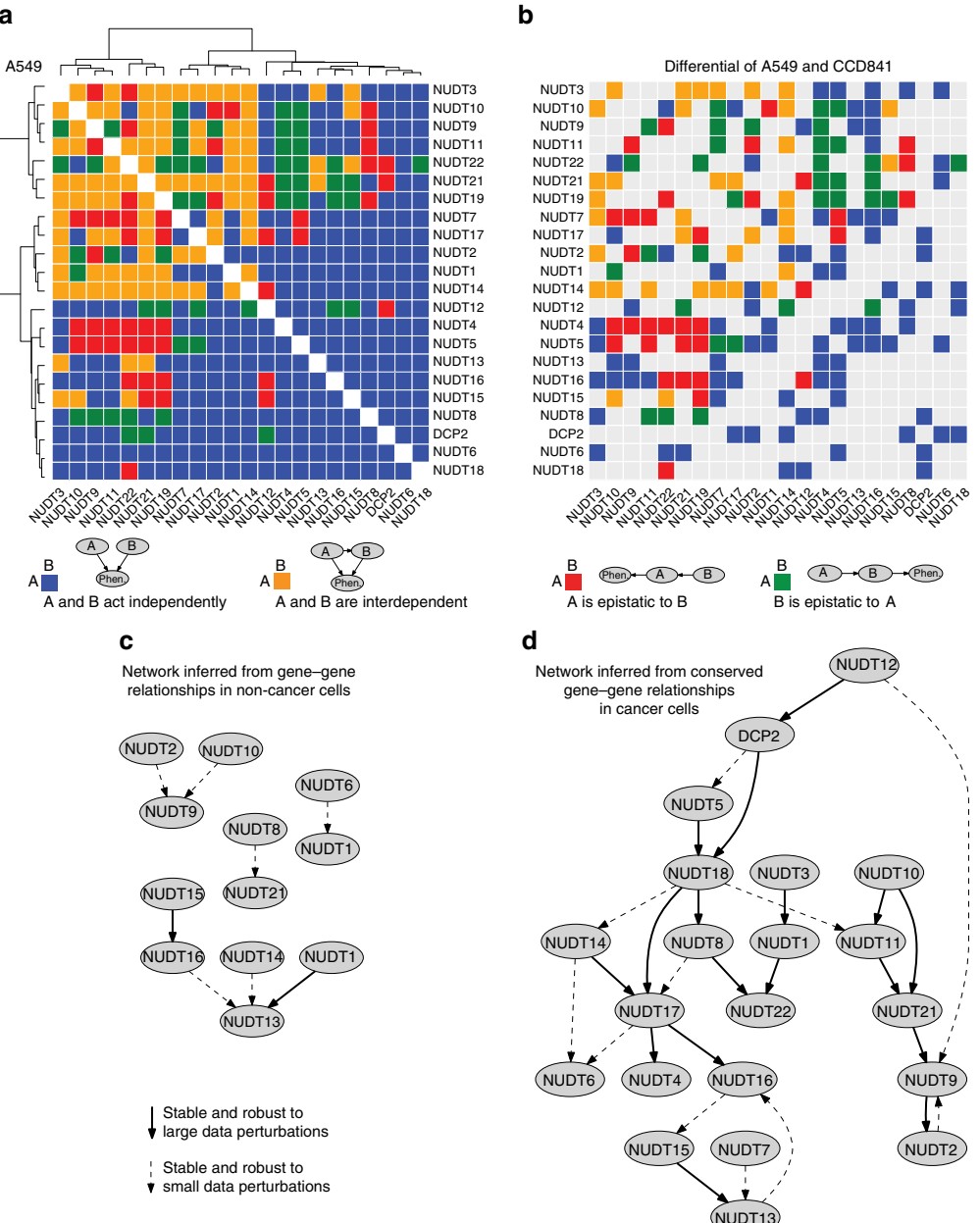

**Fig. 7** Probabilistic scoring of epistatic relationships from genetic interaction data and gene network inference. **a** Gene–gene relationships estimated from A549 cell viability data. **b** Gene–gene relationships in A549 viability data that are different from those in CCD841 viability data. **c** Gene network inferred based on gene-gene relationships in CCD841. **d** Gene network inferred based on gene–gene relationships that are conserved across A549, SW480, and MCF7. Probabilities of the estimated relationships are provided in Supplementary Fig. 6

the same degree of robustness to model parameters as solid edges, but which were more sensitive to noise added to the data. Here we show a NUDIX cancer epistasis network, importantly, with predicted directionality.

**Integrative clustering of NUDIX enzymes by data FUSION.** Given the diverse and comprehensive nature of the data sets generated and collected in this study, we aimed at conducting an integrative analysis to investigate whether the members of the human NUDIX family naturally cluster. In order to do so, we used FUSION, a recent computational method that detects clusters by fusing many different types of data measurements[19]. In short, this approach infers the so-called data latent model to create connections across heterogeneous data measurements such

as gene and protein expression profiles, substrate activity data, and genetic interaction information, and thereby extracts integrated NUDIX data profiles (see Methods section). Altogether we used 27 data sets that included measurements of 16 different types of objects (Supplementary Table 2), which we represented in an abstract scheme also known as a fusion graph[19]. We performed three *in silico* experiments in which we analyzed an entire data collection from A549, SW480, and MCF7 cells (27 data sets), and two other collections that focused specifically on data from A549 or MCF7 cells (subset of 11 data sets) (Supplementary Fig. 16).

To understand the NUDIX enzymes family at a sub-group level, we used FUSION to hierarchically cluster the data profiles extracted from the latent models of A549, SW480, and MCF7 data (Fig. 8a). To relate the clusters of the NUDIX enzymes

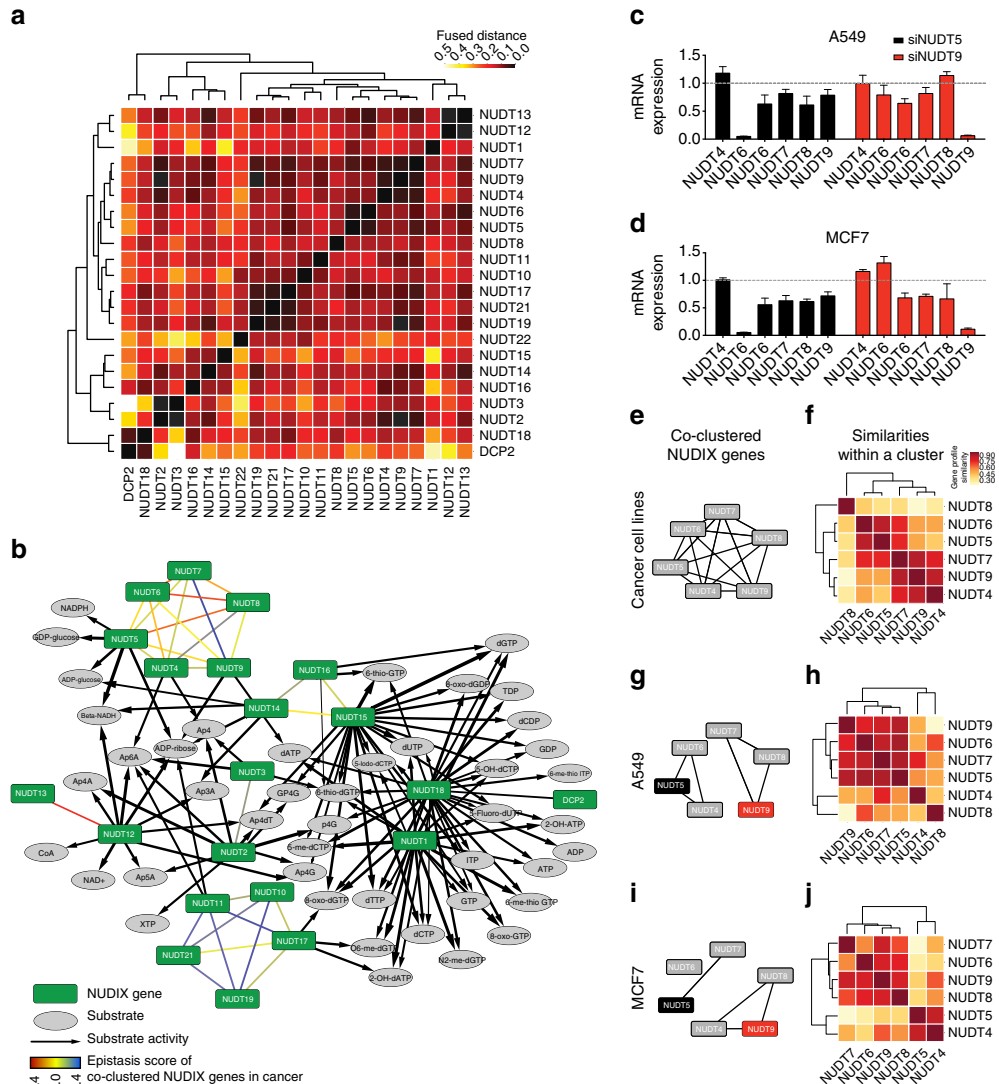

**Fig. 8** Clustering of the human NUDIX family. Integrative analysis of 27 data sets from public data repositories, such as TCGA and the HPA, as well as experimental data (Supplementary Fig. 6). **a** Detailed hierarchy of NUDIX clusters represented with a dendrogram and a heat map. Shown are distances between vector representations of NUDIX enzymes. **b** Integrative clustering analysis of the NUDIX enzymes. Enzymes in the same cluster are linked with undirected edges in the network, colored based on the epistasis score. The substrate activity data is added to the same network, relating clusters of NUDIX enzymes. **c**, **d** Effect of siRNA-mediated knockdowns of NUDT5 and NUDT9 on the mRNA expression levels of the interrogated NUDIX genes in A549 and MCF7 cells. **e** Cluster of NUDT4, NUDT5, NUDT6, NUDT7, NUDT8, and NUDT9, which is the largest identified by the integrated analysis. **f** Similarities between members of the interrogated cluster reveal internal structure of the cluster. Darker color indicates greater similarity. **g**, **h** When integrating 11 data sets that were related in A549 cell line, NUDT4, NUDT5, and NUDT6 were clustered together, but placed in a different cluster than NUDT7, NUDT8, and NUDT9. Heat map showing similarity of vector representations of the enzymes, whereby these representations were derived from the model of 11 data sets describing A549 data. **i**, **j** Similar to **g**, **h**, but in this case 10 data sets originated from MCF7 cells were considered for the clustering

identified by FUSION with the substrate activity data, we visualized the clusters together with the substrate activity data in the same network (Fig. 8b). We validated the results from the FUSION analysis by interrogating the most prominent cluster containing NUDT4, NUDT5, NUDT6, NUDT7, NUDT8, and NUDT9. We siRNA depleted NUDT5 and NUDT9 in both A549 and MCF7 cells, and evaluated the effect on expression of the rest of the NUDIX enzymes present in the cluster by qPCR (Fig. 8c, d). In both A549 and MCF7 cells, depletion of NUDT5 resulted in decreased expression of *NUDT6*, *NUDT7*, *NUDT8*, and *NUDT9*, but not *NUDT4*. This was mostly in line with the predicted FUSION clustering, which determined that the NUDIX enzymes in this group had sufficiently similar data profiles to be assigned to the same cluster (Fig. 8b). However, depletion of NUDT9 in

A549 and MCF7 resulted in a different expression pattern of the rest of the members of the cluster in the two different cell lines.

Prompted by these differences and the evidence of the non-random clustering of the NUDIX enzymes, we then performed the FUSION analysis on the separate A549 and MCF7 data sets (as opposed to the initially fused data profiles of A549, SW480, and MCF7). Interestingly, NUDT4, NUDT5, NUDT6, NUDT7, NUDT8, and NUDT9 were assigned to the same cluster when considering data from the three cancer cell lines together (Fig. 8e, f); however, when examining data collections limited to A549 (Fig. 8g, h) or MCF7 (Fig. 8i, j), these enzymes were assigned to two or three separate clusters, respectively. In A549 cells, NUDT5, NUDT6, NUDT7, and NUDT9 formed a cohesive group and were most similar to each other within the cluster

(Fig. 8h). This was confirmed by the decrease in expression of these NUDIX enzymes in both NUDT5 and NUDT9 knockdowns in A549 cells (Fig. 8c). On the other hand, *NUDT4* and *NUDT8* appeared to be unaffected or slightly increased by the depletion of NUDT9, which coincided with their outlying position in the clustering (Fig. 8c, h). Interestingly, in MCF7 cells, NUDT8 appeared to be very similar to NUDT6, NUDT7, and NUDT9 (Fig. 8j), in line with their decreased expression upon NUDT5 and NUDT9 depletion (Fig. 8d), whereas NUDT9 knockdown, resulted in an increase in *NUDT5* expression (Fig. 8b).

Seemingly then, the differences in gene expression patterns upon NUDT5 and NUDT9 depletion, the first resulting in a general decrease in the sampled NUDIX enzymes, whereas the latter resulting in increased *NUDT5* expression specifically in MCF7 cells, points towards a potential universal and non-cell-specific regulatory role of NUDT5 and a cell-specific compensatory role for NUDT9.

Given the large and diverse, yet comprehensive, amount of data that we have generated, and in order to facilitate the extraction of specific data by the scientific community, we are providing a detailed results summary (Supplementary Fig. 17).

## Discussion

The complexity of the NUDIX hydrolase activity is one of subtle nuance. Most NUDIX family proteins are easily aligned, both via sequence and structural comparisons, using both the NUDIX motif and surrounding NUDIX fold domain. In slight contrast with what has been previously reported[55], albeit with some exceptions, we found that structural and sequence alignment of the human NUDIX hydrolases can on occasion correlate with similar substrate activity[25]. Our sequence alignment analysis (Fig. 1a) identifies an expected clustering within the DIPP subfamily, which was anticipated given the large degree of similarity among these members. It was previously shown that NUDT15 had activity towards 8-oxo-dGTP[56], which prompted the authors to name NUDT15 as MTH2 due to its similarities with MTH1. However, despite possessing similar sequence and structure, detailed structural and substrate kinetics comparisons, revealing key differences in residues responsible for substrate binding, as well as distinct enzymatic activity were reported[21].

Overall, the degree of conservation among the human NUDIX enzymes resides mostly in the NUDIX box as expected; however, the substrate specificity is to a great deal determined by the rest of the protein structure.

For our activity screen of the 18 purified NUDIX enzymes, we consistently used a physiological pH, differently from what has been done before[55,57,58]. Under these conditions, most NUDIX proteins showed activity ranging from one to many substrates, confirming the activities of MTH1, NUDT2, and NUDT15 (Fig. 2a, b and Supplementary Fig. 2). As previously reported, many of the NUDIX enzymes are rather promiscuous regarding substrate preference; however, this preference was rather spread within groups of substrates (Fig. 2b). NUDT5 and NUDT14 exhibit homodimerization, share activity towards ADP-ribose and ADP-glucose, and have a high grade of structural similarity, illustrated in our structural analysis and structure superposition (Fig. 1a). We also show that NUDT14 has activity against β-NADH. Finally, NUDT22 lacks a conventional NUDIX box (Fig. 1c), which could explain its lack of activity toward any of the screened substrates. Our data show that relatively modest sequence divergence can drastically affect substrate activity and protein function.

Our global expression analysis, which has not previously been done for this family of enzymes, depicted a clear diversity of expression levels depending on the tissue of origin as well as its corresponding cancer tissue. Seemingly, in adrenal-, endometrium-, and lung-related cancers, the NUDIX enzymes were significantly highly expressed, whereas it was the opposite in kidney- and testis-related cancers (Fig. 3a). Remarkably, two distinct clusters appeared when comparing the NUDIX expression in normal and in their corresponding cancer tissues (Supplementary Fig. 3a). Similarly, the NUDIX enzymes also clustered in two groups when comparing their expression in only normal tissues (Supplementary Fig. 3b). *NUDT1*, *NUDT5*, and *NUDT14* belong to the cluster of highly expressed NUDIX in cancer, pointing toward a potential role of these NUDIX enzymes in cancer, which has been previously proposed[41,44,59,60].

A biological function in which the NUDIX enzymes seem to have a role is cell cycle regulation[49,50,61,62]. Indeed, our global approach of NUDIX depletion in a small panel of cell lines also showed that knockdown of several NUDIX enzymes altered the cell cycle distribution and affected viability, mainly in cancer cells. The cell cycle distribution of non-cancerous cells was less affected, which might be due to their slower cell cycle progression (Figs. 4–6 and Supplementary Figs. 5–12).

One of the main questions that we intended to answer was whether the human NUDIX hydrolases form an internally regulated interconnected network. To answer this we determined gene interactions, or epistasis, an approach undertaken by many others earlier[51], by depleting all of the NUDIX enzymes in a pairwise manner (Figs. 5 and 6). Elucidating epistasis is fundamental for a better understanding of genetic pathways[63]. In this context, our results clearly indicated that several of the NUDIX have an epistatic relation measured by both cell viability and cell cycle perturbations. This observation was more dramatic in cancer cells compared to non-cancerous cells (Figs. 5 and 6, and Supplementary Figs. 5 and 6).

Nodes representing alleviating interactions suggest the presence of certain redundancies among the NUDIX hydrolases. Common parallel pairwise relationships imply the existence of buffering mechanisms in the cells[64]. Interestingly, in CCD841, A549, and MCF7 cells, we found that NUDIX with strong aggravating epistatic interactions tend to be downregulated in cancer, whereas NUDIX with alleviating interactions tend to be upregulated (Fig. 5e). This observation prompts us to speculate that less favorable interactions (aggravating), may be negatively selected in a cancer context. Remarkably, a comparison between sequence distance and substrate activity similarity with epistasis scores, also revealed that NUDIX genes with negative epistatic interactions have lower Patristic sequence distances (Supplementary Fig. 6a), as well as higher Spearman's correlation of enzymatic activity similarity (Supplementary Fig. 6c), indicating that NUDIX with aggravating interactions, which are likely to belong to different pathways, have actually a similar sequence as well as substrate activity, suggesting an overlapping biochemical function, and therefore implying that there is redundancy among the NUDIX family. As an example, NUDT5 and NUDT14 are located in the low epistasis score bins in A549, MCF7, and S480 cells, and interestingly, both share high grade of sequence similarity (Fig. 1a) and substrate activity (Fig. 2a, b, d).

To be able to identify the four types of gene pairwise relationships that are common in pathways[54], we employed Réd, a probabilistic predictive model. The obtained results (Fig. 7a and Supplementary Figs. 13 and 14) suggest that all four types of pairwise relationships appear to be present in both non-cancer (Supplementary Fig. 13a) and cancer cells (Fig. 7a and Supplementary Fig. 13b, c and f). However, in the three cancer cell lines, parallelism is the most prominent pairwise relationship (indicated in blue in the heat maps in Fig. 7a and Supplementary Fig. 13b, c and f), suggesting that many NUDIX enzymes act in independent

pathways. In this context, the effects of each depletion on cell viability are compounded independently, frequently leading to a quantitative phenotype that is near a "typical" level, determined as a function of the phenotypes of the two individual gene knockdowns[65], which is what we observed in our dataset. This is in agreement with previous studies[51] reporting that strong genetic interactions, are relatively rare events. These observations further indicate that redundant and robust genetic networks are the underlying genetic architecture of the NUDIX enzymes. Consequently, substantial phenotypes are observed only when multiple genes are depleted in a given genetic network. The strong evidences of regulatory interdependence among the NUDIX hydrolases here presented could have a direct impact on the understanding of the complexity of this family of enzymes.

To integrate all our data sets and thereby obtain an overview of the overall connectivity among the NUDIX enzymes, we applied FUSION[19] (Supplementary Table 2 and Supplementary Figs. 6–10 and 13). To validate our predictions, we performed a series of qPCR experiments, which overall recapitulated the predicted clustering of NUDIX enzymes (Fig. 8). Interestingly, however, we found clear divergent results when we broke down the most prominent NUDIX cluster found by FUSION (NUDT4, NUDT5, NUDT6, NUDT7, NUDT8, and NUDT9) by depleting NUDT5 and NUDT9 in A549 and MCF7 cells (Fig. 8), overall indicating that, despite forming a clearly clustered network, the different cancer backgrounds may rely differently on the NUDIX hydrolases. These results provide a clear insight into the intricate NUDIX network, while, at the same time, pointing to specific patterns depending on the biological context.

Altogether, our comprehensive and exhaustive analysis of the human NUDIX hydrolase family not only provides a complete overview of the relationships within this interesting family of proteins but also reveals novel insights into their substrate selectivity and biological functions. Moreover, we provide a plethora of data ranging from gene and protein expression, and substrate specificity to functional genomics, which we believe are an excellent basis for future research.

## Methods

**NUDIX enzyme production**. Complementary DNA encoding NUDT4 (DIPP2α) and NUDT9 was amplified from cDNA synthesized from RNA isolated in house from HL60 cells and subcloned into pET28a(+) (Novagen). cDNAs encoding NUDT1 (MTH1), NUDT2, NUDT7, NUDT17, and NUDT18 were codon optimized for *E. coli* expression and purchased from GeneArt (Life Technologies) and subcloned into pET28a(+) (Novagen). NUDT21 and NUDT22 cDNAs were purchased from Source BioScience and were subcloned into pET28a(+). Validation of the sequences of the expression constructs was verified by sequencing. Expression constructs of NUDT3 (aa 8–172), NUDT5, the catalytic subunit of NUDT6, NUDT10 (variant AAH50700), NUDT11 (aa 13–164), NUDT12, NUDT14, NUDT15, and NUDT16 (variant AAH31215) in pNIC28 were kind gifts from SGC Stockholm. All NUDIX proteins were expressed as N-terminally His-tagged proteins, apart from NUDT10 and NUDT11 that were C-terminally tagged, in *E. coli* BL21(DE3) R3 pRARE2 at 18 °C and were purified by the Protein Science Facility (PSF) at the Karolinska Institute, Stockholm. Briefly, the N-terminally His-tagged NUDIX proteins were purified using HisTrap HP (GE Healthcare) followed by gel filtration using HiLoad 16/60 Superdex 75 (GE Healthcare). NUDT2 was expressed in BL21 DE3 (Life Technologies) overnight at 18 °C after induction with 1 mM isopropyl β-D-1-thiogalactopyranoside and purified on HisTrap HP followed by purification using HP monoQ column (GE Healthcare). Proteins were concentrated and stored at −80 °C in storage buffer (20 mM HEPES pH 7.5, 300 mM NaCl, 10% glycerol and 0.5 mM TCEP(tris-(carboxyethyl) phosphine)). Purity of protein preparations was examined using SDS-polyacrylamide gel electrophoresis followed by Coomassie staining and the mass of the purified proteins was verified using mass spectrometry.

**Activity assay**. To be able to detect both low and high activities of the NUDIX hydrolases, we performed the assay at two enzyme concentrations, low (5 nM) and high (200 nM) with 25 or 50 μM substrate, respectively (when available) (Supplementary Fig. 2b, c). NUDIX protein activity with a panel of possible substrates was assessed in technical triplicates, in reaction buffer (100 mM Tris Acetate pH 7.5, 40 mM NaCl, 10 mM MgAc, and 1 mM dithiothreitol) at 22 °C. 50 μM of the

respective substrate was incubated together with 0, 5, or 200 nM NUDIX protein diluted in reaction buffer, either without coupling enzyme, or with *E. coli* pyrophosphatase (PPase) (0.2 U ml⁻¹) or alkaline phosphatase from bovine intestinal mucosa (10 U ml⁻¹) (Sigma Aldrich), in order to detect inorganic phosphate (Pi) produced from the reaction products pyrophosphate and sugar-5-phosphates, respectively (Supplementary Fig. 2b). After 30 min incubation under shaking conditions, the generated Pi was detected by addition of a malachite green reagent[36]. After 15 min incubation, the reaction was stopped by addition of 10 μl 0.4 M sodium citrate to the 40 μl reaction mixture per well in a 384-well plate and the absorbance was read at 630 nm in an EnVision plate reader (Perkin Elmer). We normalized the absorbance signal to the control in the absence of coupling enzymes. In some cases, such as MTH1 activity toward 8-oxo-dGTP, the low concentration of enzyme was sufficient to completely convert the substrate into product, providing a maximum signal (Supplementary Fig. 2c).

**Cell cultures**. A549, MCF7, SW480, and CCD841 cells were obtained from the ATCC. All cell lines were cultured and maintained at 37 °C, with 5% $CO_2$, in Dulbecco's modified Eagle's medium (DMEM) (Invitrogen) medium, supplemented with 10% fetal calf serum (Invitrogen) and 1% penicillin/streptomycin (Sigma Aldrich). All cell lines were tested for Mycoplasma using the Mycoplasma Detection Kit (Lonza).

**siRNA-mediated knockdown**. A custom-made siRNA siGENOME (Dharmacon, see Supplementary Table 5 for sequences) library of four sequences per NUDIX enzyme was used to construct the double knockdown siRNA library. First, a sub-library containing a pool of the four sequences was built by mixing equal amounts of each sequence, diluted to a final concentration of 20 μM using a Janus Automated Workstation (PerkinElmer). Second, the pooled sub-library was used to generate the 276 pairwise combinations using an Echo550 Liquid Handler (Labcyte) to add 60 nl of each siRNA pool to BD Falcon 384-well plates. Before seeding of the cells, Opti-MEM medium was added to each well. Transfection agent Lipofectamine RNAiMAX (Life Technologies) was diluted in Opti-MEM medium (Life Technologies) and added to the wells. The siRNA-RNAiMAX mix was incubated for 25–30 min at room temperature (RT), and finally 30 μl of cell suspension was added per well. The final concentration of siRNA was 18 nM (9 nM per pool). The non-targeting siRNA 5 (Dharmacon) was used as negative control, as well as load equivalent control to compare single vs double knockdown. As positive transfection controls siRNA against KIF11, PLK1, and UBB (Dharmacon) were used, as the depletion of these essential genes causes cell death within 48 h, indicating a successful transfection. The double knockdown experiments were done in technical triplicates, whereas the single knockdown experiments were done in two technical triplicate sets. For statistics refer to Supplementary Fig. 4.

**Multiple alignment and phylogenetic analysis**. The protein coding sequences of the Human NUDIX family proteins were aligned in the ClustalOmega software[22]. Both, full-length and NUDIX fold domain sequences were aligned separately and used in subsequent phylogenetic analysis. MrBayes (v3.2)[23] was used to construct Bayesian inference trees with Markov chain Monte Carlo methods. Tree generation was performed using the Blosum62 model[66], indicated as the best-fit model after a mixed model fitting test in MrBayes, using unconstrained branch length priors with no root enforcement. After an initial burn-in of the first 25% of trees, samples were analyzed every 1,000 generations and convergence assessed by the average SD of split frequencies (<0.01). The posterior probabilities for internal nodes of the consensus tree were calculated from the posterior density of trees.

**NUDIX structure visualization**. Known structures of NUDIX family proteins were generated in Pymol (The PyMOL Molecular Graphics System, Version 1.8 Schrödinger, LLC). Individual structures are shown in cartoon format with the NUDIX box colored in neon blue, the NUDIX domain colored in smudge green, and any remaining structure colored in gray. Superpositions of phylogenetic relevant structures were generated in Pymol with the NUDIX box in cartoon format, colored neon blue, and the remaining structures in ribbon format with complimentary colors.

**Expression analysis**. We downloaded the RNA-seq V2 level 3 data for all cancer types from the TCGA Data Portal and extracted quantile normalized gene expression levels. The RNA sequencing data of normal healthy tissues from the HPA project[40] were processed from raw reads according to the TCGA V2 pipeline as described in the document "TCGA mRNA-seq Pipeline for UNC data": (https://www.cghub.ucsc.edu/docs/tcga/UNC_mRNAseq_summary.pdf), using the same versions of all software and references, and the same normalization scheme. Finally, data for the NUDIX family genes were extracted from the expression matrices and used for further analyses. Log2-transformed expression values for normal tissues were used for hierarchical clustering and visualized in a heat map. For tumors, the fold change compared with the corresponding normal tissues (according to Supplementary Table 1) was used for clustering. Significantly up- or downregulated NUDIX genes in the normal-cancer comparisons were determined using two-sample *t*-tests, with false discovery rate-adjusted *p*-values. This was illustrated in a bubble plot where the radius of each bubble is proportional to the

square root of the expression level, and the significance level and direction of change is shown in the color of the bubble.

**Cell cycle analysis**. Image-based cell cycle analysis was performed as follows[46]: upon siRNA knockdown in 384-well plates, images of Hoechst 33342 (Sigma-Aldrich)-stained cells were acquired using a high-throughput microscope. The segmentation of the nuclei and measure of integrated intensity was done by Cell Profiler image analysis software[67], and the cell cycle distribution was calculated and plotted using PopulationProfiler[46].

**Tissue microarrays**. The TMAs consisted of the standard set of 44 normal tissues and 20 different forms of cancer used in the HPA project[40], and were constructed as follows[68]: TMAs were constructed from formalin-fixed paraffin-embedded normal tissues and tumors obtained from the pathology archives at the Uppsala Akademiska Hospital, Uppsala, Sweden, with permission from the Research Ethics Committee at Uppsala University (2002-577, 2009/139 and 2011/473). For normal tissues, one 1.0 mm diameter core from each donor block representing three individuals per normal tissue was used. For cancer tissues, duplicate 1.0 mm diameter cores from up to 12 individuals per cancer type were used. The donor cores were assembled into recipient TMA blocks using the Beecher Manual Tissue Arrayer MTA-1 (Estigen OÜ, Tartu, Estonia). All blocks were cut in 4 μm-thick sections using waterfall microtomes (Microm HM 355S, Thermo Fisher Scientific, Fremont, CA, USA), dried overnight at RT and baked in 50 °C for 12–24 h before immunohistochemical staining.

**Immunhistochemistry**. Immunohistochemical procedures followed the standard techniques employed by the HPA project[68]. In short, buffers and instrumentation were purchased from LabVision (Fremont, CA, USA), where not stated otherwise. TMA slides were deparaffinized and hydrated using a histostaining instrument (Leica Autostainer XL, Leica Microsystems, Wetzlar, Germany). The deparaffinization was performed in Xylene (Histolab, Gothenburg, Sweden) twice for 15 min, followed by hydration in graded alcohols, including blocking of endogenous peroxidase activity by 0.3% $H_2O_2$ in 95% ethanol for 5 min. After deparaffinization, the slides were immersed in retrieval citrate buffer, pH 6, and for heat-induced epitope retrieval a pressure boiler (Decloaking chamber, Biocare Medical, Walnut Creek, CA, USA) was used at 125 °C for 4 min. Automated immunohistochemical staining of TMA slides was performed in Autostainer 480 as previously described[68]. Slides were rinsed at RT in wash buffer supplemented with 0.2 % Tween-20 between each step. Primary antibodies (Supplementary Table 4) were diluted in antibody diluent and allowed to incubate for 30 min at RT. Subsequently, slides were incubated with enhancer reagent for 20 min, followed by UltraVision LP HRP polymer for 30 min before being developed using DAB QUANTO for 5 min. After staining, the tissues were counterstained with Mayer's hematoxylin (Histolab) for 7.5 min, dehydrated in ethanol and Tissue-Clear Xylene substitute (Sakura Finetek, Alphen aan den Rijn, The Netherlands), mounted using Pertex (Histolab), and coverslipped by the Leica Autostainer XL. All tissues were scanned using an automated slide-scanning system with a ×20 objective (ScanScope XT, Aperion Technologies, Vista, CA, USA) and manually evaluated by experienced pathologists.

**Gene silencing and qRT-PCR**. siRNA complexes were prepared containing 2.5 nmol siRNA per 6 μL Lipofectamine 2000 per sample, in DMEM with 2% fetal bovine serum without antibiotics, incubated for 30 min at RT, to allow complex formation, and loaded into 6-well plates. Subsequently, $7 \times 10^4$ cells per well were seeded on the siRNA/Lipofectamine 2000 mix. A549 and MCF7 cells were siRNA transfected for 96 h. siGENOME siRNA (Dharmacon) was used for siRNA-mediated depletion (Supplementary Table 3). Lipofectamine 2000 (Invitrogen) was used as transfection agent in accordance with the manufacturer's instructions (see above). After the incubation, total RNA was isolated using GeneJet RNA Purification kit (Thermo Scientific). Four hundred nanograms of mRNA was reverse transcribed using Maxima First Strand cDNA Synthesis Kit for RT-qPCR with dsDNase (Thermo Scientific). Real-time quantitative PCR (qRT-PCR) was performed using the iTaq Universal SYBR Green Supermix (BioRad), 5 ng cDNA and primers on a CFX96 qRT-PCR machine (BioRad). The primers used for qRT-PCR can be found in Supplementary Table 5. Relative mRNA levels were calculated using the iQ5 Optical System Software (BioRad) and using hGAPDH and β-actin as reference housekeeping genes.

**Production of *E. coli* PPase**. cDNA encoding *E. coli* PPase was amplified from genomic *E. coli* DNA using PCR and subcloned into pNIC28. *E. coli* PPase was expressed in *E. coli* BL21(DE3) R3 pRARE2 at 18 °C and purified using HisTrap HP (GE Healthcare) followed by gel filtration using HiLoad 16/60 Superdex 75 (GE Healthcare) by PSF.

**Scoring genetic interactions**. Under our definition, a genetic interaction was assigned to a pair of genes $(a,b)$ if $W_{ab}$, the cell viability of the double knockdown, was significantly different from $E(W_{ab})$, the viability of the double knockdown that would be expected if $a$ and $b$ were noninteracting. The expected viability $E(W_{ab})$

predicted for a strain mutated in genes $a$ and $b$ was quantified according to the multiplicative neutrality function. For each gene pair, the difference between the means of $W_{ab}$ and $E(W_{ab})$ was assessed by using a $Z$-test with a significance level $\alpha$ = 0.05. Interactions were classified as synergistic (aggravating or negative) if $W_{ab} < E(W_{ab})$ and alleviating (positive) if $W_{ab} > E(W_{ab})$. Significant negative and positive interactions were shown with red and blue edges, respectively, connecting the corresponding genes.

**Gene network inference (Réd)**. Cell viability measurements of single and double NUDIX knockdowns from three experiments were independently normalized to the negative siRNA controls. Afterwards, the normalized values were averaged. The multiplicative rule[52] was used to compute the expected cell viability of double knockdowns under the assumption that corresponding single knockdowns do not interact. Réd[53] was used to automatically infer gene networks from knockdown data. Inputs to the algorithm of Réd were a matrix of double knockdown viability measurements ($G$), a vector of single knockdown viability measurements ($S$), and a matrix of expected cell viability ($H$). The Réd inference algorithm involved three stages. (i) first, the algorithm factorized matrix $G$. A factorized model was represented with two latent matrices that captured the global structure of the cell viability landscape and accounted for potential noise in the data. The latent dimension (factorization rank) was selected as a value at which the best reconstruction of matrix $G$ was obtained measured as normalized root-mean-square error. The model also inferred a logistic map that represented a nonlinear mapping from latent to cell viability space. This nonlinear transformation took into consideration differences in single knockdown backgrounds, which affected cell viability of double knockdowns. (ii) In the second stage, Réd applied a probabilistic scoring scheme to the inferred model with the goal of estimating probabilities of gene-gene relationships of four different types. Given genes $u$ and $v$, Réd distinguishes between four possible pairwise relationships: (1) $u$ and $v$ act in a linear pathway, where $v$ is epistatic to $u$; (2) $u$ and $v$ act in a linear pathway, where $u$ is downstream of $v$; (3) $u$ and $v$ affect the phenotype separately; and (4) $u$ and $v$ are partially interdependent. Réd provided probabilistic estimates of the possible pairwise relationships for all combinations of NUDIX genes. The estimates were visualized in a heat map, where the four types of gene-gene relationships were color coded, and used for hierarchical clustering based on the Hamming distance between relationship-based gene profiles. (iii) In the third stage, Réd algorithm used the pairwise scores to construct a gene network. Stability of the network was evaluated against independent and normally distributed noise added to the data, whose mean was set to zero and standard deviation was varied between $10^{-6}$ and a third of the SD of $G$. For every noise level value, $M = 100$ independent runs of Réd algorithm were executed. Network edges were categorized into two groups, where edges that appeared in at least 40% of the runs were visualized with solid lines and edges that appeared between 20% and 40% of the runs were visualized with dashed lines.

**Data fusion by collective matrix factorization (FUSION)**. We performed three data fusion experiments, each considering a different data collection. In the largest experiment we analyzed 27 data sets describing relations between objects of 16 different data types and taking into consideration all NUDIX related data in A549, SW480, and MCF7 cells (Supplementary Fig. 16). The other experiments independently analyzed data about one cell line at a time, either A549 or MCF7. As such, these experiments considered subsets of the entire data collection showed in Supplementary Fig. 16. In particular, the A549 experiment used 11 data sets and the MCF7 experiment used 10 data sets.

The experiments adopted the following computational procedure: first, each data set was represented with a data matrix. This was possible because each data set was considered as a relation between objects of two different types. For example, cell cycle data for A549 cells were encoded in two types of matrices. One matrix had NUDIX genes in rows and cell cycle phases in columns and its elements contained data about the number of single NUDIX knockdown cells in the corresponding cell cycle phases. The five other matrices had NUDIX genes both in rows and columns. Given a cell cycle phase, a matrix contained data about the number of double NUDIX knockdown cells detected in respective cell phase. Second, data sets were organized in a relational map called data fusion graph[19]. Third, data matrices were separately normalized in two steps by first dividing each row by its second norm and then similarly dividing each column. Fourth, collective matrix factorization algorithm was applied to jointly co-factorize all data matrices considered in the experiment. The algorithm compressed each data matrix through a tri-factorization, which estimated three latent matrices for a given data matrix. The latent matrices had substantially fewer dimensions than the data matrix but, when multiplied together, they provided a high quality reconstruction of the data matrix. Given a data matrix $R_{ij}$, the algorithm approximated it with a product $R_{ij} \approx G_i S_{ij} G_j^T$, where $G_i$, $S_{ij}$ and $G_j$ are the inferred latent matrices. The strategy used to select the number of latent dimensions (factorization rank) is described elsewhere[69]. The essence of data fusion was to reuse the latent data matrices when co-factorizing data matrices that reported on objects of common data type. This property of the algorithm was central for data fusion and necessary to achieve transfer of information between data sets considered in each experiment. Finally, a collection of inferred latent matrices were analyzed further to obtain data profiles of the NUDIX genes.

**Clustering NUDIX proteins using the estimated latent space**. Collective matrix factorization algorithm (FUSION) provided a system of latent matrices that were used to compute profiles (feature vectors) of the NUDIX genes. The latent matrices were multiplied resulting in non-sparse, smooth, and complete reconstruction of input data matrices, which were originally sparse, non-smooth, and, as such, inappropriate for clustering. The reconstructed matrices, which reported on NUDIX genes, were used to compare NUDIX genes to each other. Given two NUDIX genes and their feature vectors returned by the fusion algorithm, the cosine distance between feature vectors was calculated. The cosine distance compared two feature vectors on a normalized space by computing the cosine of the angle between the vectors. This setup generated one gene-gene distance matrix for each data matrix and contained pairwise distances between NUDIX genes. Distance matrices were averaged resulting in a single distance matrix that integrated distances from all data sets. The resulting distance matrix was used for hierarchical clustering and for visualization.

Additional information on the methodologies used can be found in Supplementary Note 1.

**Code availability**. The respective codes for FUSION and Réd are available at GitHub:

FUSION—http://github.com/marinkaz/scikit-fusion
Réd—http://github.com/biolab/red

**Data availability**. The data that support the findings of this study are available from the corresponding author upon request.

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

## Acknowledgements

This work has been supported by the European Union–Marie Curie-FP7-People programme under grant agreement number 332428, as well as the Torsten and Ragnar Söderberg Foundation, and the Knut and Alice Wallenberg Foundation. Further support was received from the Swedish Research Council, the European Research Council, Swedish Cancer Society, the Swedish Children's Cancer Foundation, AFA insurance, the Swedish Pain Relief Foundation, The Cancer Society in Stockholm, the Wenner-Gren Foundations, the Göran Gustafsson Foundation, Vinnova and the Swedish Foundation for Strategic Research. We thank Professor Barry Potter and Andrew Riley from the Department of Pharmacology of the Oxford University, for kindly providing 5-PP-InsP5.

## Author contributions

J.C.-P. and T.H. designed the research and wrote the manuscript. T.H. supervised the project. J.C.-P. coordinated the collaborations, performed the siRNA, cell cycle, and substrate activity assays together with A.-S.J., M.H. and U.M. M.Z. designed and performed the bioinformatics analysis. B.H. performed the analysis on expression data. M.C. performed the structure analysis. A.-S.J., O.L. and Z.K. produced the NUDIX proteins used in the study. J.E.U. performed the DIPPs substrate-specific experiments and knockdown validation by qRT-PCR. J.M.C.-M. performed the qPCR experiments. C.L. and P.-H.E. performed the protein analysis on TMAs. D.M. and C.W. provided the informatics tool for the cell cycle analysis. R.P.A.B. contributed to the structure analysis and consensus tree construction. M.S. provided some bioinformatics insights. B.L., E.L.L.S., E.L., P.S. and B.Z. contributed to the supervision of their corresponding sections.

## Additional information

**Competing interests:** The authors declare no competing financial interests.

