## [Peer Review File · Nature Communications]

Reviewers' comments:

Reviewer #1 (Remarks to the Author):

This manuscript on the human NUDIX hydrolases, including the well-studied enzyme MTH1, addresses an important topic in cancer research, among other research areas. Members of this research team have advanced the importance of these enzymes recently. The “study design” for the work described here was a survey-type investigation of various properties of known human NUDIX class enzymes. Thus, a quantitative, convincing demonstration of properties of any one enzyme was not revealed, in spite of a lengthy Supp. Section. Instead, a qualitative assessment of various NUDIX family features, such as substrate specificity (Fig 2), mRNA/protein expression (Fig. 3), cellular viability & cell cycle response to NUDIX enzyme depletion in four cell lines (Figs. 4, 5), and a genetic interaction analysis in the four cell lines studied (Figs. 6, 7,). The interrelationship analyses in Figures 7 and 8 are interesting. Overall, the analysis presented is novel and provides useful information in the mode of “hypothesis generation,” essentially as pointed out by the authors at the very end of the text.

Minor points and other comments:

1. Figure 1 is appropriate as a portion of Supp. Figure 1, but is confusing to a reader and is not especially useful as Figure 1 in the main text.
2. The legend of Figure 3 needs to have some corrections.
3. One element that is lacking in the text, is a set of comments on the limitations of interpretation of the approaches used. This concern applied to all of the methods and database analyses described. The manuscript would be enhanced significantly with addition of such material.

Reviewer #2 (Remarks to the Author):

This paper represents an extremely ambitious and detailed analysis of the structures, expression and functions of the members of the human nudix hydrolase family. It uses both original experimental data and data extracted from online databases and relies heavily on single and double gene knockdown experiments. It seeks to explore proposed “interconnected genetic networks” and a “global biochemical role” for these hydrolases in order to explain their functions. Ultimately, however, these functions are not explained much beyond what is understood already, with the accumulated data serving mainly as a resource and basis for further

work, which nevertheless could be valuable.

A number of general points require consideration by the authors.

1. It is not clear why the authors expect there to be any “interconnected genetic networks” or a “global biochemical role” for the nudix hydrolases rather than specific and individual well-defined functions. This could be better explained.

2. The nudix hydrolases are well known for their overlapping substrate specificities and promiscuity. In lines 525-7 the authors state that “This limited promiscuity may indicate that the NUDIX enzymes, despite being spread across species, have more specific biological functions than previously anticipated”. While it is probably true that the different nudix hydrolases do have specific functions, the authors have probably underestimated the level of substrate promiscuity within the family through choosing a single set of assay conditions that may not reflect the microenvironments of the various enzymes and through omitting certain substrates, e.g. full-length capped RNAs (“mCAP structure” is presumably a cap analog?), inorganic polyphosphates, acyl-CoAs and non-nucleotide polyphosphates that can be used by several nudix hydrolases. Also, the data in Figure 2a do not fully agree with previously published work, while many other as yet undocumented examples of overlap involving untested or obscure nucleotide and non-nucleotide substrates are likely to exist. For this reason, it is possible that the proposed epistasis is in many cases more artificial than real. When hydrolase A with a particular specificity is knocked down, then another hydrolase B with an overlapping specificity may partially compensate but this situation would only arise and be of any significance in the knocked down cells. B may have no role to play in A’s pathway in the normal cell but could have its normal (unconnected) role compromised by the competing burden of A’s substrate(s) in the knockdown. Thus it would be premature to talk of true “genetic interactions” or of hydrolases being in the “same or different pathway”.

3. To some extent, this possibility could be addressed by discussing whether the proposed networks and interactions make any sense on the basis of existing knowledge about substrate specificities and proposed functions of the nudix hydrolases. Throughout the paper, there is insufficient recognition of the large body of literature pertaining to nudix hydrolases. Indeed, there is evidence of a lack of familiarity. More needs to be done in this regard.

4. For the original experimental data, there is no indication of the reproducibility of the experiments e.g. how many times were the knockdowns performed? The scale of the experiments does not obviate the need for repetition and appropriate statistical analysis. As much of the analysis is reliant on the robustness of these data, this point must be addressed.

5. No data are provided to indicate the success and extent of the knockdowns in reducing protein

levels. Where antibodies are available, Western blots should be provided as a supplemental figure to indicate the degree of protein knockdown achieved. As hydrolases are catalysts, profound depletion may be required to have the necessary effect on the level of the relevant substrate. This possibility should be acknowledged. Where antibodies are not available qRT-PCR data should be provided as a substitute.

6. The quality of the English in the main text is variable, from excellent to vague and confusing. Some examples are provided below but there are many others. The paper would benefit from being edited by a native English speaker who is familiar with the work.

Further specific points for consideration and revision (numbers refer to line number in merged file):

67: MTH1 was called MTH1 long before the NUDT1 nomenclature was proposed. One nomenclature or the other should be used consistently in the manuscript and figures e.g. Supp. Fig. 1 MTH1, Supp. Fig. 2 NUDT1 etc.

73-75 Several other labs have made major contributions to the characterization of nudix hydrolases. They also deserve credit, not just Bessman, whose albeit excellent work is almost exclusively on bacterial enzymes.

82 a reference or references should be provided to support the general statement that “nudix enzymes are upregulated following cellular stress”. Most published data refers to plants and may not be relevant to humans.

94 briefly describe what FUSION is

113-5 the similarities between NUDT7&8 and NUDT12&13 have previously been pointed out in the literature

116 “phosphate” should be “polyphosphate”

122-3 (see also 720-1) previously published data show that NUDT19 shares significant sequence similarity with NUDT7 and shares substrate preference (Ofman, 2006, BJ 393, 537). The insert between the nudix box and the UPF0035 motif potentially involved in CoA binding leads to misalignment in multiple alignments.

141 It would be helpful to know why only 18 of the 22 hydrolases were studied. Was expression of the others unsuccessful?

156-72 Ap4, Ap4G and Gp4 are not novel substrates for NUDT2-type enzymes. They have previously been described. Ap4dT is entirely predictable based on the lack of base and sugar specificity of this enzyme (see Guranowski, 2000, Pharm. Ther. 87, 117). Ap6A is not a novel NUDT3 substrate (Safrany, 1999, JBC 274, 21735). The substrate overlap of NUDT12 with NUDT2&5 has previously been noted (Abdelraheim, 2003, BJ 374, 329) (see also lines 520-2). Most of the reported substrates for NUDT14 have also previously been described (Yagi, 2003, BJ 370, 409). NUDT7 has been well characterized before with substrates used here. Is lack of activity here due to inactive recombinant enzyme? Lack of activity of NUDT6 is consistent with

attempts of Murphy et al. to find substrates while lack of activity of NUDT21 and 22 is probably due to highly mutated or loss of nudix box. NUDT21 does bind but does not hydrolyze Ap4A (Yang, 2010, PNAS 107, 10062). Greater acknowledgment of previous work is required.

170 should be Supplemental Figure 2C, not 1C

176 “affinity” is confusing. Better to use “similarity”. Affinity is also misused in Supp. Fig. 3. “Usage” or “utilization” would be better.

176-8 it is not immediately clear how substrate activity correlations can be reliably included in Fig. 2B for NUDT6 and NUDT21 when no significant activity has been detected with any substrate used (Fig. 2A). This should be clarified. As indicated above, NUDT21 is unlikely to have a substrate.

182 the data in Supp. Fig. 3 are impossible to understand without much more explanation of the experiments, their aims and conclusions. The legend is inadequate e.g it is not even mentioned that CCD841 etc. are cell lines.

221-40 again, a greater attempt should be made to compare data to previously published results (where known) which agree or differ in order to validate experiment e.g, NUDT5 previously reported to be expressed in all tissues

258-9 it would be helpful to highlight how specific knockdown of NUDT 10 and 11 was achieved given their almost identical gene sequences — were UTRs used exclusively?

255-6 not clear what is meant by “a wide range of cell cycle effects”

266 G1 accumulation in NUDT13-depleted CCD841 cells is not evident in Fig. 4B “in concordance with the survival data” for these cells.

271-2 these lines are poorly written. Should be “... despite the lack of known biological roles for some of the nudix hydrolases, depletion experiments indicate they may be essential for ...”

Further, in most cases can they be truly described as “essential” rather than “normally required”?

273 (and elsewhere, e.g. lines 348, 539) observed cell cycle perturbations upon nudix hydrolase knockdown do not necessarily imply direct roles in cell cycle regulation (e.g. as with cyclin/CDKs etc.) as is stated. The effects could be secondary to true regulation due to nucleotide pool imbalances and resulting toxicity such as DNA strand breaks.

279-80 replace “display a” with “equate to”

296 the choice of a multiplicative rather than additive model for assessing epistasis should be justified briefly

415-55 “Figure 7” should read “Figure 8”

483-6 clustering of the members of the DIPP subfamily is not really “new”. They are already known to be closely related, which the authors even acknowledge themselves.

1223-7 Panels C and D in the legend to Fig. 3 are wrongly referred to as B and C

Supp. Figures

63 why is “cell cycle” in the title to Supp Fig. 2?

Unsigned

Reviewer #3 (Remarks to the Author):

The authors of the manuscript titled "A comprehensive structural, biochemical and biological profiling of the human NUDIX hydrolase family" submitted to Nature Communications have performed a multifaceted study of the human representative of the NUDIX superfamily of enzymes, surveying complementing aspects, both bioinformatic and experimental of these enzymes. Such comprehensive surveys, attempting to assign biologically relevant functional features to families of enzymes, taking large context into account are very useful for the community. They report the details of mRNA and protein expression in multiple cell types, as well as measurements of the overlapping or unique involvement in pathways related to the cell cycle and cancer. Hence, this compendium of knowledge can be used in the future to reach deeper understanding of the involvement of these proteins in key cellular processes, with promising pharmaceutical potential. In general, the findings presented here are interesting to the wide readership of Nature Communications. Nevertheless, there are several points that still need to be re-visited and clarified in order to support the claims made in this study.

Major comments

1. [lines 106, 712-713, 725-727] The method used for generation of the NUDIX phylogenetic model is unclear. Jukes-Cantor model refers to nucleotides, rather than amino acids. It is not clear which server/software/stand alone method was used for tree generation (all details of the methodology need to be reported). In addition, the tree shown in Figure 1 shows quite a few cases of polytomy, and I wonder if the authors can try to resolve this in order to obtain a more robust tree (this is true also for the large 605*NUDIX tree; Sup. Figure 1A). For 22 sequences with such abundance of structural data, a maximum-likelihood/Bayesian method for tree generation can be applied, with higher chances of obtaining a tree with higher robustness than the one obtained by neighbor joining. In addition, as far as I understand, the large (605 NUDIX sequences) and the small (22 human NUDIX sequences) alignments are based on full-length sequences; A comparison between the full-length sequence alignment, and another alignment based on the NUDIX-only domain sequence alignment will reveal if the NUDIX classes presented here are indeed robust (this can be assessed by comparing the full vs. domain MSA-based trees). In lines 174-175, for instance, NUDT2 and NUDT5 are presented as an example of two enzymes that belong to the same class, but demonstrating different substrate affinity. Looking at the alignment in Sup. Figure 1A, these two enzymes look as if the percent identity

between them is pretty low. It seems as if the human NUDIX paralogs, in general, are very diverse, hence naturally hard to group into distinct sets. Another example is NUD5/6/9, that belong to the same structural class, and are also tightly clustered based on the FUSION analysis. I believe that supporting the structure-based grouping of the human NUDIX proteins will help a lot in establishing one of the main claims that appear in the abstract – that substrate selectivity of NUDIX enzymes does not correlate with structural classes.

2. Useful data dissemination. This paper was submitted as a “resource” type of publication. Indeed, the authors have carried out an impressive array of assays and associated analysis, having significant impact on the field. Nevertheless, readers that are interested in carrying out deeper analysis of NUDIX enzymes may find it frustrating to orient themselves in the massive amount of reported data. Indeed, the authors have used the FUSION algorithm to locate the low hanging fruits based on the multiples types of data, but still, in order for the results to be more accessible, I believe that writing a summary (supplementary text) for each human NUDIX, containing the main finding and most significant inter-relations with other NUDIX enzymes will be of great benefit. An alternative way to present the data is creating a table that compares previous knowledge and the major new insight gained for each human NUDIX enzyme.

Clarifications

1. [lines 114-115] - The term “high grade of similarity” should be better explained – for instance by reporting percent identity, and comparing this number to the average percent identity between the human NUDIX sequences.
2. [lines 120-122] - The similarity between NUDT16 and NUDT20, as shown in the 22-member tree is not in accord with the relationship between these two NUDIX enzymes in the large tree (Sup. Figure 1A). Such inconsistencies should be reported (Are there additional such cases?)
3. [lines 158-159] The overlap between NUDT12/2/5 is not clearly seen in Figure 2A.
4. [line 177] – It is not clear what was compared using the spearman correlation (“between sequence and substrate activity”) – what measure was used to compare the sequence/structure similarity?
5. [line 215] – Please refer here to the correlation between the mRNA expression levels and the protein expression levels in Cancer cells. Reporting the correlation between these trends will allow assessing the confidence in these results (For NUDIX proteins for which mRNA & protein levels are available for the same cell/cancer type).
6. [lines 740-742] – Please describe what are the values that provided the log₂-transformed ratio in the normal cell mRNA expression levels.
7. Supplementary figure 3 legend [line 71] the term “structural sequence similarity” is not clear. What was measured here?
8. Figure 8B portrays the network of FUSION-based epistasis scores as well as substrate activity. Some of the relationships there are not clear – maybe due to the layout algorithm used for displaying the network. For instance, cases like substrate activity arrows connecting between substrates (GP4G-Ap4dT) are unclear.

9. In line 480, reference 36 is mentioned as claiming that sequence similarity is a good proxy for functional similarity; the abstract of the referred paper states that "...Although these three human proteins (NUD1/15/8) resemble each other in their sequences, their substrate specificities differ considerably". Please clarify.
10. In lines 524-527, it is not clear how the results reported in this paragraph lead to the conclusion that the phylogenetically diverse NUDIX enzymes have more specific biological functions than anticipated.
11. Please refer to the limitations of the methodologies utilized in this research. For instance, the possible effect of NUDIX alternative splicing on mRNA/protein expression profiles, and the biological (eukaryotic, human) relevance of the screened substrates (you may consult the notions reported in the recent NUDIX substrate screening paper, Nguyen VN et al., *Proteins*. 2016 Sep 12, PMID: 27618147).

Minor

1. [all page footnotes] Wrong journal name.
2. [line 46] The term "multiple alignment heat maps" is not common enough to be used in the abstract.
3. [line 94] Missing reference to the FUSION algorithm.
4. [line 102] "Structural overlap" is a vague term; maybe structural similarity? (also in line 471).
5. [line 126] The term "motif" commonly refers to a short sequence signature, and not a globular domain. The term "NUDIX domain" may be a better fit.
6. [line 170] Reference to supplemental figure should be to 2C (rather than 1C)
7. [line 323] Reference to Supplemental Figure 12A is unclear; could not find such a panel in that figure.
8. [line 345] Reference to Supplemental Figure 12B should be 14B?
9. [line 365] Comma can be omitted.
10. [line 414] Reference to Supplementary figure 13 should be 15 (?)
11. [line 417-455,618-637] Wrong reference to Fig 7 instead of 8
12. [lines 456-461] Please consider splitting the lengthy sentence into two parts.
13. [lines 502-527,601-617] Please try to shorten the phrasing and relate to the proposed meaning of the results, rather than repeating what is already written in the results. Phrases such as "In other words, for instance" can be improved.
14. [line 679] E Coli->E. coli
15. [Line 838] E.Coli->E. coli
16. [line 1184] In figure 2B legend, it is not clear what is "structure distance" – is it RMSD? Or maybe the branching probability in the tree?
17. [line 1223] Fig. 3 legend – panel B is referred to, rather than panel C.
18. [line 1227] Fig. 3 legend – panel C is referred to, rather than panel D.
19. [line 1272] Fig. 6B: the color of the alleviating interaction seems more cyan/blue than green (see network node color...). Also, it is unclear what the width of the network edges stand for

(statistical significance?). Please consider moving panel A to the supplemental figures, since the message of this analysis is sufficiently demonstrated in panel A.

20. [Supplementary Figure 1] NUDT19 and NUDT18 appear twice in the tree, and DCP2, NUDT8 and NUDT9 are missing.

21. [Supplementary Table 2] spreadsheet name is “antibodies”.

22. Please use a single term for the relationships between NUDIX enzymes and their substrates – substrate affinity/activity/preference/specificity/selectivity are used throughout (e.g. line 419 etc); I assume “activity” can be used throughout.

Reviewer #4 (Remarks to the Author):

The manuscript of Puigvert et al. reports the first comprehensive NUDIX enzyme profiling map using systematic *in vitro* and *in silico* investigations in order to understand their biological functionality. For the *in vitro* tests: (1) They have screened 18 of the human NUDIX proteins in the presence of 52 different substrates to elucidate their biochemical redundancies. (2) In order to connect biochemical and biological functions they siRNA depleted all individual NUDIX proteins and checked how their depletion affects cell viability and cell cycle. (3) In order to elucidate interdependence among the enzymes, the authors also depleted them in pairwise to generate epistatic interaction map. Moreover, using the analytical tool RéD they were able to reconstruct the detailed pathway structures of the enzymes.

For the *in silico* analysis: (1) The authors have determined the structure-sequence overlap of the NUDIX enzymes and grouped them according to their similarity. (2) In order to compare the NUDIX hydrolyses gene expression patterns in healthy and cancer tissues they compared the gene/protein expression of the NUDIX enzymes in different cancer tissues using the Cancer Genome Atlas database. (3) As a last step using their comprehensive data set they performed an integrative analysis to investigate whether the members of the human NUDIX family naturally cluster.

Their major findings are: (1) Structural and sequence alignment of the Human NUDIX hydrolyses does not always predict similar substrate activity. (2) NUDIX enzymes have promiscuous preference for several substrates, however this promiscuity was rather limited, indicating that the NUDIX enzymes have more specific biological functions. (3) Their global expression analysis revealed a clear diversity in the expression level, however they were able to depict specific NUDIX hydrolyses that can be essential for cancer cells. (4) They also showed that several NUDIX knockdowns altered the cell cycle and affected viability mainly in cancer cells. (5) Alleviating was the most common pairwise relationship between NUDIX hydrolyses which suggests the presence of certain redundancy and robustness. (6) They found that the genetic architecture of the NUDIX enzymes comprised from the genetic networks are redundant and robust. (7) Integrating all their data sets and thereby obtaining an overview of the overall

connectivity among the NUDIX allowed them to estimate the extent to which the expression of one NUDIX gene is dependent on another NUDIX gene. Overall the paper provided some novel insight into the relationships among these proteins and provided a comprehensive dataset for future research.

Points that need attention:

The authors claim an overlap of the genetic interactions among the three cancer cell lines (FIGURE 4) or among the cancer cell lines and normal tissue. Can the authors provide a statistical test for this overlap? Can they plot a scatter plot of the epistasis score of all interacting pairs in one cell line versus another and test for correlation?

To identify significant interactions between each gene pair, the difference between the means of W_{ab} and $E(W_{ab})$ was assessed by using a Z test with a significance level $\alpha=0.1$. How are the results affected if a more strict cut-off value for alpha is used (like 0.05 or 0.01)?

It is not clear to me how the calculation was done for the spearman rank correlation between structure distances and substrate activity measurements for each pairwise NUDIX enzymes.

It would be important to show which are the common gene-gene relationships derived from the three different cancer cell lines that also diverge from those identified in normal non-cancer cells (FIGURE 7 and Supplementary Figure 6).

More generally, I feel that there is a wealth of data but that the overall message is not as strong. Much more in depth analysis of the data can be performed. For example:

Based on the overlap of the substrate specificity of the NUDIX enzyme pairs can the authors predict the genetic interactions (positive/negative) between them in normal tissue?

Do NUDIX enzymes with frequent epistatic interactions (identified in the normal cells) tend to change their RNA/protein expression level (compared to the normal tissue) more frequently in different cancer cell lines?

Based on their large-scale genetic interaction and gene-gene relationship data can the authors predict one or two potentially good target pairs from the NUDIX enzymes for intervention, e.g., treatment of the different cancer cell lines?

Reviewer #5 (Remarks to the Author):

In the submitted manuscript, Puigvert et al. report an in-depth analysis of the human NUDIX genes, a family of hydrolases with mostly unknown biological function. The authors have integrated different datasets from own experiments and public databases to create a comprehensive NUDIX gene map, aiming at providing an insight into the biological functions of and possible redundancies between NUDIX enzymes.

The authors thoroughly describe their analyses and visualize their findings in very complex figures. They have collected an impressive dataset and use it to cluster the human NUDIX genes. The methods applied seem to be carefully chosen. However, the overall findings of the paper stay a little behind the expectations that the abstract raises. In general, text and figures are hard to read and to understand and it is difficult to extract the findings and judge the methods, especially if someone has a different research background.

Specific points:

1. The text is in parts hard to read and some sentences are confusing and need multiple readings in order to get the content. The language is sometimes sloppy and there are sentences with grammar mistakes or missing words. Some examples are found in lines 385, 529-536, see also points 8 and 9. The manuscript should be carefully edited and checked for spelling mistakes and comprehensibility.
2. The figures are too complex and some of them are very unintuitive (especially, but not limited to, Figures 2b, 3a, 3c, 6c, 8b). The manuscript would benefit from condensing the figures and trying to present important differences (e.g. Figures 3a, 3c). Showing all data available is well meant, but does not help and some information could be moved to the supplements.
3. In lines 314-316, it is claimed that the three cancer cell lines used share a substantial amount of genetic interactions, while the respective figure (5C) shows the exact opposite. Some overlap is always there and is to be expected by chance. Most of the interactions found are specific to the cell lines.
4. The statistics behind the epistasis analyses described in results and methods remain somewhat unclear. It should be clearly stated how many replicates were made, what the significance cut-off of each test was (p values in the figures!) and at what false discovery rate the interactions were called.
5. Along that line, it seems that in Figures 5A and 6D not only significant interactions are shown, but all deviations from the expected value are depicted in color code, giving the visual impression that almost all genes share interactions. As mentioned in the text, true interactions are rare and an interaction matrix is rather sparse, especially when considering such a small set of genes.

6. The cell cycle based genetic interactions shown for the cell cycle phenotypes show a clear difference between the non-cancer cell line (many/mostly increase in cell fraction) and the other cell lines (mostly decrease in cell fraction). This difference can also be an analysis artifact and is not properly discussed. Also, in cell line A549, NUDT5 suddenly shares an increase interaction with almost all others, but all decrease interactions are gone.
7. Title of Figure 2 names information about cell cycle regulation, which is not found in the figure.
8. Legend of Figure 3 names figures A-B, B, C instead of A-B, C, D.
9. Legend of Figure 5: It should be ‘A genetic interaction was assigned to pairS of genes if the viability phenotype_ of the double knockdown...’
10. The paper lacks a proper comparison between all the different clusters of NUDIX genes that were found with different methods. The differences between structure and enzyme activity clusters are discussed, but in the end the reader is left to put together all the information. The information given in Figure 8 is very comprehensive, but in itself lacks hints to shared or different biological properties of the NUDIX genes/proteins. While condensing some of the figures, an additional summary of the found clusters and their differences to the clusters that came out of each dataset would be of benefit.
11. In order to really deduct function, it would be a good approach to see how the NUDIX genes cluster within a larger set of genes, not which clusters form in the family. Of course, a large-scale epistasis analysis is beyond the scope of this paper, but for most of the analyzed databases, information is available for huge numbers of genes representing different biological functions and processes. Clustering of NUDIX genes to different processes based on the comprehensive data analyzed could really point to biological function and redundancies.

Stockholm 15 May 2017

To: Nature Communications senior editor Stéphane Larochelle

Dear editor,

Please, find below the point by point answers to the referees' comments to our manuscript entitled "A comprehensive structural, biochemical and biological profiling of the human NUDIX hydrolase family" with tracking number **NCOMMS-16-24225A**.

We have substantially revised the manuscript, addressing all the referee's concerns, and we have re-submitted the manuscript to Nature Communications.

Yours sincerely,

Jordi Carreras-Puigvert

Point by point answers to reviewers' comments:

Reviewer #1 (Remarks to the Author):

This manuscript on the human NUDIX hydrolases, including the well-studied enzyme MTH1, addresses an important topic in cancer research, among other research areas. Members of this research team have advanced the importance of these enzymes recently. The “study design” for the work described here was a survey-type investigation of various properties of known human NUDIX class enzymes. Thus, a quantitative, convincing demonstration of properties of any one enzyme was not revealed, in spite of a lengthy Supp. Section. Instead, a qualitative assessment of various NUDIX family features, such as substrate specificity (Fig 2), mRNA/protein expression (Fig. 3), cellular viability & cell cycle response to NUDIX enzyme depletion in four cell lines (Figs. 4, 5), and a genetic interaction analysis in the four cell lines studied (Figs. 6, 7,). The interrelationship analyses in Figures 7 and 8 are interesting. Overall, the analysis presented is novel and provides useful information in the mode of “hypothesis generation,” essentially as pointed out by the authors at the very end of the text.

Minor points and other comments:

1. Figure 1 is appropriate as a portion of Supp. Figure 1, but is confusing to a reader and is not especially useful as Figure 1 in the main text.

> A new Figure 1 has been generated, which should be of great use to understand the global structure similarities within the human NUDIX enzymes.

2. The legend of Figure 3 needs to have some corrections.

> Corrections have been made to the legend of Figure 3.

3. One element that is lacking in the text, is a set of comments on the limitations of interpretation of the approaches used. This concern applied to all of the methods and database analyses described. The manuscript would be enhanced significantly with addition of such material.

> We have added an extensive paragraph commenting the limitations of the methods used in our study.

Reviewer #2 (Remarks to the Author):

This paper represents an extremely ambitious and detailed analysis of the structures, expression and functions of the members of the human nudix hydrolase family. It uses both original experimental data and data extracted from online databases and relies heavily on single and double gene knockdown experiments. It seeks to explore proposed “interconnected genetic networks” and a “global biochemical role” for these hydrolases in order to explain their functions. Ultimately, however, these functions are not explained much beyond what is understood already, with the accumulated data serving mainly as a resource and basis for further work, which nevertheless could be valuable.

A number of general points require consideration by the authors.

1. It is not clear why the authors expect there to be any “interconnected genetic networks” or a “global biochemical role” for the nudix hydrolases rather than specific and individual well-defined functions. This could be better explained.

> Answer to the reviewer: the NUDIX hydrolases have been extensively characterized, however often in non-physiological conditions, and certainly not always have the human enzymes have been studied. Given that the NUDIX hydrolases form a large family of enzymes that seem to be structurally similar, clustering in several classes, and also seem to share similar substrates, we hypothesized that if there were compensatory responses within the same family of enzymes, it could be in an interconnected manner. We acknowledge that some of the NUDIX have individually well-defined functions, however given the overlap among the substrates of some of the family members, it is plausible, as we are able to show in our epistatic networks, that an interconnected genetic network exists.

2. The nudix hydrolases are well known for their overlapping substrate specificities and promiscuity. In lines 525-7 the authors state that “This limited promiscuity may indicate that the NUDIX enzymes, despite being spread across species, have more specific biological functions than previously anticipated”. While it is probably true that the different nudix hydrolases do have specific functions, the authors have probably underestimated the level of substrate promiscuity within the family through choosing a single set of assay conditions that may not reflect the microenvironments of the various enzymes and through omitting certain substrates, e.g. full-length capped RNAs (“mCAP structure” is presumably a cap analog?), inorganic polyphosphates, acyl-CoAs and

non-nucleotide polyphosphates that can be used by several nudix hydrolases. Also, the data in Figure 2a do not fully agree with previously published work, while many other as yet undocumented examples of overlap involving untested or obscure nucleotide and non-nucleotide substrates are likely to exist. For this reason, it is possible that the proposed epistasis is in many cases more artificial than real. When hydrolase A with a particular specificity is knocked down, then another hydrolase B with an overlapping specificity may partially compensate but this situation would only arise and be of any significance in the knocked down cells. B may have no role to play in A's pathway in the normal cell but could have its normal (unconnected) role compromised by the competing burden of A's substrate(s) in the knockdown. Thus it would be premature to talk of true "genetic interactions" or of hydrolases being in the "same or different pathway".

> Answer to the reviewer: we agree that the conditions in which we screened the potential substrates are limiting the number of substrates that could be identified, however, compared to other studies, our conditions were chosen to be closer to be physiologically relevant, maintaining a constant pH of 7.5. That said, the hypothesis of a genetic interaction is on the basis of the substrate promiscuity of some of the family members. In order to provide enough evidences for an epistatic relationship at a hydrolase/metabolic level, experiments showing the levels of the products or metabolites of the hydrolysis reactions carried out by the different NUDIX upon their respective knockdowns, would ultimately elucidate the involvement of each NUDIX in the corresponding pathways. However, these experiments, namely HPLC or radiolabeling, were out of reach. Instead, we interrogated the hypothesis by generating a matrix of 276 double knockdowns, which allowed us to interpolate potential epistatic genetic relationships using the algorithm RéD. These interactions are clearly represented by the survival phenotype as well as cell cycle regulation, in which single versus double knockdown indicate, in each case, that some NUDIX belong to the same or different pathway, respectively.

3. To some extent, this possibility could be addressed by discussing whether the proposed networks and interactions make any sense on the basis of existing knowledge about substrate specificities and proposed functions of the nudix hydrolases. Throughout the paper, there is insufficient recognition of the large body of literature pertaining to nudix hydrolases. Indeed, there is evidence of a lack of familiarity. More needs to be done in this regard.

> We agree with the referee that there is a large body of papers describing the NUDIX proteins that should be cited. We have therefore extended our reference list to better acknowledge the vast amount of

work that has been performed in the field. An extensive comparison of the data presented in the manuscript and previously published data has also been added in the results and discussion.

4. For the original experimental data, there is no indication of the reproducibility of the experiments e.g. how many times were the knockdowns performed? The scale of the experiments does not obviate the need for repetition and appropriate statistical analysis. As much of the analysis is reliant on the robustness of these data, this point must be addressed.

> A clarification has been added to the corresponding Methods section and an additional Supplemental figure showing the high correlation between the two runs of single siRNA knockdowns has been added.

5. No data are provided to indicate the success and extent of the knockdowns in reducing protein levels. Where antibodies are available, Western blots should be provided as a supplemental figure to indicate the degree of protein knockdown achieved. As hydrolases are catalysts, profound depletion may be required to have the necessary effect on the level of the relevant substrate. This possibility should be acknowledged. Where antibodies are not available qRT-PCR data should be provided as a substitute.

> A new supplemental figure has been generated in which knockdown efficiency for all NUDIX enzymes, with the exception of NUDT22 is shown by qRT-PCR.

6. The quality of the English in the main text is variable, from excellent to vague and confusing. Some examples are provided below but there are many others. The paper would benefit from being edited by a native English speaker who is familiar with the work.

> We apologize for this, and have revised the text accordingly.

Further specific points for consideration and revision (numbers refer to line number in merged file):

67: MTH1 was called MTH1 long before the NUDT1 nomenclature was proposed. One nomenclature or the other should be used consistently in the manuscript and figures e.g. Supp. Fig. 1 MTH1, Supp. Fig. 2 NUDT1 etc.

> MTH1 is used in the manuscript when we refer to the protein, while NUDT1 is used when referring to the gene.

73-75 Several other labs have made major contributions to the characterization of nudix hydrolases. They also deserve credit, not just Bessman, whose albeit excellent work is almost exclusively on bacterial enzymes.

> We agree with the reviewer that other labs besides Bessman deserve credit for their work on NUDIX hydrolases and we acknowledge that we failed to refer to these labs appropriately earlier. In fact, there is a vast number of papers contributing to the knowledge about the NUDIX hydrolase superfamily and it is impossible to reference them all. We therefore decided to refer to the excellent reviews written by Bessman and McLennan on NUDIX hydrolases in the introduction and refer to other labs contributions when we compare our data to others later in the manuscript. A number of new references describing work on NUDIX hydrolases have been added.

82 a reference or references should be provided to support the general statement that “nudix enzymes are upregulated following cellular stress”. Most published data refers to plants and may not be relevant to humans.

> These references have been added: Increase of MTH1 levels in various tumors, Okamoto et al., 1996, Iida et al., 2001, Kennedy et al., 2002. Increase of NUDT5 expression after treatment with H₂O₂, Nakayama et al., 2010.

94 briefly describe what FUSION is

> An overview of the method is added to section “Integrative clustering of NUDIX enzymes by data FUSION”. More details are found in “Methods.”

113-5 the similarities between NUDT7&8 and NUDT12&13 have previously been pointed out in the literature

> We thank the reviewer for this comment, however we couldn't find existing literature comparing human NUDT7 and NUDT8 as well as NUDT12 and NUDT13 enzymes. Nonetheless, we have clarified our findings regarding these enzymes in the text.

116 “phosphate” should be “polyphosphate”

> Corrected

122-3 (see also 720-1) previously published data show that NUDT19 shares significant sequence similarity with NUDT7 and shares substrate

preference (Ofman, 2006, BJ 393, 537). The insert between the nudix box and the UPF0035 motif potentially involved in CoA binding leads to misalignment in multiple alignments.

> We agree with the reviewer that structural and substrate similarities have been previously reported by Ofman, Gasmi and Reilly, who reported these similarities mainly in the mouse enzymes. Our analysis indicated a partial structural similarity between the human NUDT7 and NUDT19, however the absence of purified NUDT19 and the lack of activity of our purified NUDT7 over CoA, made this comparison not possible. Nevertheless, a discussion has now been added in the text.

141 It would be helpful to know why only 18 of the 22 hydrolases were studied. Was expression of the others unsuccessful?

> The reason for this has now been clarified in the text. Indeed, the expression of 4 of the 22 hydrolases was unsuccessful.

156-72 Ap4, Ap4G and Gp4 are not novel substrates for NUDT2–type enzymes. They have previously been described. Ap4dT is entirely predictable based on the lack of base and sugar specificity of this enzyme (see Guranowski, 2000, Pharm. Ther. 87, 117). Ap6A is not a novel NUDT3 substrate (Safrany, 1999, JBC 274, 21735). The substrate overlap of NUDT12 with NUDT2&5 has previously been noted (Abdelraheim, 2003, BJ 374, 329) (see also lines 520-2). Most of the reported substrates for NUDT14 have also previously been described (Yagi, 2003, BJ 370, 409). NUDT7 has been well characterized before with substrates used here. Is lack of activity here due to inactive recombinant enzyme? Lack of activity of NUDT6 is consistent with attempts of Murphy et al. to find substrates while lack of activity of NUDT21 and 22 is probably due to highly mutated or loss of nudix box. NUDT21 does bind but does not hydrolyze Ap4A (Yang, 2010, PNAS 107, 10062). Greater acknowledgment of previous work is required.

> The text has now been revised for clarification.

170 should be Supplemental Figure 2C, not 1C

> This has been corrected

176 “affinity” is confusing. Better to use “similarity”. Affinity is also misused in Supp. Fig. 3. “Usage” or “utilization” would be better.

> Changed to activity and utilization

176-8 it is not immediately clear how substrate activity correlations can

be reliably included in Fig. 2B for NUDT6 and NUDT21 when no significant activity has been detected with any substrate used (Fig. 2A). This should be clarified. As indicated above, NUDT21 is unlikely to have a substrate.

> We agree with the reviewer, the NUDIX enzymes for which no activity was detected have been removed from the analysis.

182 the data in Supp. Fig. 3 are impossible to understand without much more explanation of the experiments, their aims and conclusions. The legend is inadequate e.g it is not even mentioned that CCD841 etc. are cell lines.

> A new figure has been made for better clarification and explanation.

221-40 again, a greater attempt should be made to compare data to previously published results (where known) which agree or differ in order to validate experiment e.g, NUDT5 previously reported to be expressed in all tissues

> We acknowledge the lack of comparison in the previous version of the manuscript, this has been amended in the current version.

258-9 it would be helpful to highlight how specific knockdown of NUDT 10 and 11 was achieved given their almost identical gene sequences — were UTRs used exclusively?

> We acknowledge the reviewer's comment and agree that the specificity of the siRNA against NUDT10 and NUDT11 is not high enough to distinguish each gene given the high grade of sequence similarity. This has been noted in the text.

255-6 not clear what is meant by “a wide range of cell cycle effects”

> This has now been explained and text has been rephrased.

266 G1 accumulation in NUDT13-depleted CCD841 cells is not evident in Fig. 4B “in concordance with the survival data” for these cells.

> This has been explained in the rephrased text.

271-2 these lines are poorly written. Should be “... despite the lack of known biological roles for some of the nudix hydrolases, depletion experiments indicate they may be essential for ...” Further, in most cases can they be truly described as “essential” rather than “normally required”?

> This has been explained in the rephrased text.

273 (and elsewhere, e.g. lines 348, 539) observed cell cycle perturbations upon nudix hydrolase knockdown do not necessarily imply direct roles in cell cycle regulation (e.g. as with cyclin/CDKs etc.) as is stated. The effects could be secondary to true regulation due to nucleotide pool imbalances and resulting toxicity such as DNA strand breaks.

> Explained and rephrased in the text

279-80 replace “display a” with “equate to”

> Replaced

296 the choice of a multiplicative rather than additive model for assessing epistasis should be justified briefly

> This has been clarified in section “Survival and cell cycle genetic interaction networks of NUDIX enzymes uncover biological redundancies”. Briefly, previous work (Mani et al. PNAS 2008) provided multiple lines of evidence favoring the multiplicative genetic interaction model for studies seeking to identify functional relationships.

415-55 “Figure 7” should read “Figure 8”

> Corrected

483-6 clustering of the members of the DIPP subfamily is not really “new”. They are already known to be closely related, which the authors even acknowledge themselves.

> Clarified and corrected

1223-7 Panels C and D in the legend to Fig. 3 are wrongly referred to as B and C

> Corrected

Supp. Figures

63 why is “cell cycle” in the title to Supp Fig. 2?

> Corrected, mistake due to different document version.

Unsigned

Reviewer #3 (Remarks to the Author):

The authors of the manuscript titled "A comprehensive structural, biochemical and biological profiling of the human NUDIX hydrolase family" submitted to Nature Communications have performed a multifaceted study of the human representative of the NUDIX superfamily of enzymes, surveying complementing aspects, both bioinformatic and experimental of these enzymes. Such comprehensive surveys, attempting to assign biologically relevant functional features to families of enzymes, taking large context into account are very useful for the community. They report the details of mRNA and protein expression in multiple cell types, as well as measurements of the overlapping or unique involvement in pathways related to the cell cycle and cancer. Hence, this compendium of knowledge can be used in the future to reach deeper understanding of the involvement of these proteins in key cellular processes, with promising pharmaceutical potential. In general, the findings presented here are interesting to the wide readership of Nature Communications. Nevertheless, there are several points that still need to be re-visited and clarified in order to support the claims made in this study.

Major comments

1. [lines 106, 712-713, 725-727] The method used for generation of the NUDIX phylogenetic model is unclear. Jukes-Cantor model refers to nucleotides, rather than amino acids. It is not clear which server/software/stand alone method was used for tree generation (all details of the methodology need to be reported). In addition, the tree shown in Figure 1 shows quite a few cases of polytomy, and I wonder if the authors can try to resolve this in order to obtain a more robust tree (this is true also for the large 605*NUDIX tree; Sup. Figure 1A). For 22 sequences with such abundance of structural data, a maximum-likelihood/Bayesian method for tree generation can be applied, with higher chances of obtaining a tree with higher robustness than the one obtained by neighbor joining. In addition, as far as I understand, the large (605 NUDIX sequences) and the small (22 human NUDIX sequences) alignments are based on full-length sequences; A comparison between the full-length sequence alignment, and another alignment based on the NUDIX-only domain sequence alignment will reveal if the NUDIX classes presented here are indeed robust (this can be assessed by comparing the full vs. domain MSA-based trees). In

lines 174-175, for instance, NUDT2 and NUDT5 are presented as an example of two enzymes that belong to the same class, but demonstrating different substrate affinity. Looking at the alignment in Sup. Figure 1A, these two enzymes look as if the percent identity between them is pretty low. It seems as if the human NUDIX paralogs, in general, are very diverse, hence naturally hard to group into distinct sets. Another example is NUD5/6/9, that belong to the same structural class, and are also tightly clustered based on the FUSION analysis. I believe that supporting the structure-based grouping of the human NUDIX proteins will help a lot in establishing one of the main claims that appear in the abstract – that substrate selectivity of NUDIX enzymes does not correlate with structural classes.

> We are very grateful for the reviewer's comment. Indeed, the reviewer has brought to our attention a better method for tree generation. Our revised manuscript now includes a phylogenetic tree (Figure 1A and Supplemental Figure 1B) built using MrBayes implementation of Bayesian inference trees with Markov chain Monte Carlo (MCMC) methods. We have extensively tested and changed the approach used for the sequence analysis used in our study, which is reflected in extensive change in the text as well as new and more clarifying figures.

2. Useful data dissemination. This paper was submitted as a “resource” type of publication. Indeed, the authors have carried out an impressive array of assays and associated analysis, having significant impact on the field. Nevertheless, readers that are interested in carrying out deeper analysis of NUDIX enzymes may find it frustrating to orient themselves in the massive amount of reported data. Indeed, the authors have used the FUSION algorithm to locate the low hanging fruits based on the multiples types of data, but still, in order for the results to be more accessible, I believe that writing a summary (supplementary text) for each human NUDIX, containing the main finding and most significant inter-relations with other NUDIX enzymes will be of great benefit. An alternative way to present the data is creating a table that compares previous knowledge and the major new insight gained for each human NUDIX enzyme.

> We agree with the reviewer, given the large amount of data that we generated, a summary would be very helpful to the reader. Therefore, we have added an extensive summary figure (Supplemental Figure 16) in which all results are collected for each NUDIX.

Clarifications

1. [lines 114-115] - The term “high grade of similarity” should be better explained – for instance by reporting percent identity, and comparing

this number to the average percent identity between the human NUDIX sequences.

> We have included % of Pairwise identity as well as Patristic distance information in Supplemental Figure 1A for a more accurate explanation.

2. [lines 120-122] - The similarity between NUDT16 and NUDT20, as shown in the 22-member tree is not in accord with the relationship between these two NUDIX enzymes in the large tree (Sup. Figure 1A). Such inconsistencies should be reported (Are there additional such cases?)

>The text has been revised and Sup. Figure 1A is no longer included

3. [lines 158-159] The overlap between NUDT12/2/5 is not clearly seen in Figure 2A.

> This has now been clarified in the text

4. [line 177] – It is not clear what was compared using the spearman correlation (“between sequence and substrate activity”) – what measure was used to compare the sequence/structure similarity?

> We have generated new box plots that we hope will clarify this comparison. Also, the text has been revised accordingly.

5. [line 215] – Please refer here to the correlation between the mRNA expression levels and the protein expression levels in Cancer cells. Reporting the correlation between these trends will allow assessing the confidence in these results (For NUDIX proteins for which mRNA & protein levels are available for the same cell/cancer type).

> We thank the reviewer for this comment. We agree that mRNA and protein levels correlation would be very interesting and helpful for subsequent analysis. We initially considered using existing data to draw such correlations, however, after extensive search in Proteomics databases such as www.proteomicsdb.org and the Human Protein Atlas, we realized that the reliability and availability of NUDIX protein levels was insufficient, therefore we decided not to try to make an mRNA/protein level correlation. The data shown in Figure 3C is qualitative, therefore making it not possible to correlate with quantitative mRNA data for instance from the same HPA or TCGA.

6. [lines 740-742] – Please describe what are the values that provided the log2-transformed ratio in the normal cell mRNA expression levels.

> The log₂-transformed mRNA expression values for the normal tissues were obtained from the Human Protein Atlas database. These data are based on RNA-seq in units of fragments per kilobase of transcript per million fragments mapped. This has now been clarified in the text.

7. Supplementary figure 3 legend [line 71] the term “structural sequence similarity” is not clear. What was measured here?

> We have included a revised figure for clarity (now Supplemental Figure 6) as well as clarified in the revised text.

8. Figure 8B portrays the network of FUSION-based epistasis scores as well as substrate activity. Some of the relationships there are not clear – maybe due to the layout algorithm used for displaying the network. For instance, cases like substrate activity arrows connecting between substrates (GP4G-Ap4dT) are unclear.

> This figure has been revised to improve readability and reduce clutter.

9. In line 480, reference 36 is mentioned as claiming that sequence similarity is a good proxy for functional similarity; the abstract of the referred paper states that “...Although these three human proteins (NUD1/15/8) resemble each other in their sequences, their substrate specificities differ considerably”. Please clarify.

> We agree with the reviewers’ comment; the previous reference was incorrect. Our new analysis provides a better understanding of the correlation between structure and substrate specificity, which is reflected in the new version of the manuscript.

10. In lines 524-527, it is not clear how the results reported in this paragraph lead to the conclusion that the phylogenetically diverse NUDIX enzymes have more specific biological functions than anticipated.

> Rather than being largely promiscuous, we found that the promiscuity of the NUDIX enzymes over their substrates is limited to subgroups of substrates. This may imply that the reactions in which the NUDIX enzymes are involved in, are more restricted in diversity than previously thought, therefore pointing to a more specific biological function.

11. Please refer to the limitations of the methodologies utilized in this research. For instance, the possible effect of NUDIX alternative splicing on mRNA/protein expression profiles, and the biological (eukaryotic, human) relevance of the screened substrates (you may consult the

notions reported in the recent NUDIX substrate screening paper, Nguyen VN et al., Proteins. 2016 Sep 12, PMID: 27618147).

> We agree with the reviewer and have added a paragraph referring to the limitations of the methods used in our study. With special emphasis to the substrate availability.

Minor

1. [all page footnotes] Wrong journal name. > Fixed
2. [line 46] The term “multiple alignment heat maps” is not common enough to be used in the abstract. > This term is no longer used.
3. [line 94] Missing reference to the FUSION algorithm. > Added
4. [line 102] “Structural overlap” is a vague term; maybe structural similarity? (also in line 471). > Changed
5. [line 126] The term “motif” commonly refers to a short sequence signature, and not a globular domain. The term “NUDIX domain” may be a better fit. > Changed
6. [line 170] Reference to supplemental figure should be to 2C (rather than 1C) > Fixed
7. [line 323] Reference to Supplemental Figure 12A is unclear; could not find such a panel in that figure. > Fixed, it should be Suppl. Fig 13
8. [line 345] Reference to Supplemental Figure 12B should be 14B? >Yes, fixed
9. [line 365] Comma can be omitted. >Fixed
10. [line 414] Reference to Supplementary figure 13 should be 15 (?) >Yes, fixed
11. [line 417-455,618-637] Wrong reference to Fig 7 instead of 8 >Fixed
12. [lines 456-461] Please consider splitting the lengthy sentence into two parts.
13. [lines 502-527,601-617] Please try to shorten the phrasing and relate to the proposed meaning of the results, rather than repeating what is already written in the results. Phrases such as “In other words, for instance” can be improved.
14. [line 679] E Coli->E. coli >Fixed
15. [Line 838] E.Coli->E. coli >Fixed
16. [line 1184] In figure 2B legend, it is not clear what is “structure distance” – is it RMSD? Or maybe the branching probability in the tree?
17. [line 1223] Fig. 3 legend – panel B is referred to, rather than panel C.
18. [line 1227] Fig. 3 legend – panel C is referred to, rather than panel D.
19. [line 1272] Fig. 6B: the color of the alleviating interaction seems more cyan/blue than green (see network node color...). Also, it is unclear what the width of the network edges stand for (statistical significance?). Please consider moving panel A to the supplemental

figures, since the message of this analysis is sufficiently demonstrated in panel A.

> We have revised the caption for Figure 6. Also, we have removed panel C, which was unclear. However, for clarity and to be able to explain panel B, we have left panel A in the figure.

20. [Supplementary Figure 1] NUDT19 and NUDT18 appear twice in the tree, and DCP2, NUDT8 and NUDT9 are missing.

> This figure panel has been replaced for clarity.

21. [Supplementary Table 2] spreadsheet name is “antibodies”.

> Fixed

22. Please use a single term for the relationships between NUDIX enzymes and their substrates – substrate affinity/activity/preference/specificity/selectivity are used throughout (e.g. line 419 etc); I assume “activity” can be used throughout.

> We have now clarified and used only substrate activity or substrate specificity when appropriate throughout the manuscript.

Reviewer #4 (Remarks to the Author):

The manuscript of Puigvert et al. reports the first comprehensive NUDIX enzyme profiling map using systematic in vitro and in silico investigations in order to understand their biological functionality. For the in vitro tests: (1) They have screened 18 of the human NUDIX proteins in the presence of 52 different substrates to elucidate their biochemical redundancies. (2) In order to connect biochemical and biological functions they siRNA depleted all individual NUDIX proteins and checked how their depletion affects cell viability and cell cycle. (3) In order to elucidate interdependence among the enzymes, the authors also depleted them in pairwise to generate epistatic interaction map. Moreover, using the analytical tool RéD they were able to reconstruct the detailed pathway structures of the enzymes.

For the in silico analysis: (1) The authors have determined the structure-sequence overlap of the NUDIX enzymes and grouped them according to their similarity. (2) In order to compare the NUDIX hydrolyses gene expression patterns in healthy and cancer tissues they compared the gene/protein expression of the NUDIX enzymes in different cancer tissues using the Cancer Genome Atlas database. (3) As a last step using their comprehensive data set they performed an integrative

analysis to investigate whether the members of the human NUDIX family naturally cluster.

Their major findings are: (1) Structural and sequence alignment of the Human NUDIX hydrolyses does not always predict similar substrate activity. (2) NUDIX enzymes have promiscuous preference for several substrates, however this promiscuity was rather limited, indicating that the NUDIX enzymes have more specific biological functions. (3) Their global expression analysis revealed a clear diversity in the expression level, however they were able to depict specific NUDIX hydrolyses that can be essential for cancer cells. (4) They also showed that several NUDIX knockdowns altered the cell cycle and affected viability mainly in cancer cells. (5) Alleviating was the most common pairwise relationship between NUDIX hydrolyses which suggests the presence of certain redundancy and robustness. (6) They found that the genetic architecture of the NUDIX enzymes comprised from the genetic networks are redundant and robust. (7) Integrating all their data sets and thereby obtaining an overview of the overall connectivity among the NUDIX allowed them to estimate the extent to which the expression of one NUDIX gene is dependent on another NUDIX gene. Overall the paper provided some novel insight into the relationships among these proteins and provided a comprehensive dataset for future research.

Points that need attention:

The authors claim an overlap of the genetic interactions among the three cancer cell lines (FIGURE 4) or among the cancer cell lines and normal tissue. Can the authors provide a statistical test for this overlap? Can they plot a scatter plot of the epistasis score of all interacting pairs in one cell line versus another and test for correlation?

> We think that the reviewer refers to Figure 5 instead of 4. As suggested, we did a scatter plot of epistasis scores of all interacting pairs in one cell line versus another cell line. That is, we represented each NUDIX pair with a two-dimensional point, such that the first coordinate is the epistasis score of that NUDIX pair in one cell line, and the second coordinate is the epistasis score of that NUDIX pair in the other cell line. Statistical analysis of correlation confirmed that there is significant association between genetic interactions from different cell lines (A549 vs. SW480: $\rho=0.535$, $p\text{-value}=1.63 \times 10^{-18}$; A549 vs MCF7: $\rho=0.539$, $p\text{-value}=8.04 \times 10^{-19}$; SW480 vs. MCF7: $\rho=0.473$, $p\text{-value}=2.72 \times 10^{-14}$).

To identify significant interactions between each gene pair, the difference between the means of W_{ab} and $E(W_{ab})$ was assessed by using a Z test with a significance level $\alpha=0.1$. How are the results

affected if a more strict cut-off value for alpha is used (like 0.05 or 0.01)?

> We agree with the reviewer. In order to give a broader view of the significant interactions, we now provide both $\alpha=0.1$ and $\alpha=0.05$ interactions.

It is not clear to me how the calculation was done for the spearman rank correlation between structure distances and substrate activity measurements for each pairwise NUDIX enzymes.

> The method used to calculate the spearman rank correlation has been added to the text. Additionally, new box plot figures have replaced the bar plots for clarity.

It would be important to show which are the common gene-gene relationships derived from the three different cancer cell lines that also diverge from those identified in normal non-cancer cells (FIGURE 7 and Supplementary Figure 6).

> We agree with the reviewer, we have added a new panel in Supplemental Figure 6 (Now Supplemental Figure 14, panel g) indicating the differential gene-gene relationships between the cancer and non-cancer cell lines.

More generally, I feel that there is a wealth of data but that the overall message is not as strong. Much more in depth analysis of the data can be performed. For example:

Based on the overlap of the substrate specificity of the NUDIX enzyme pairs can the authors predict the genetic interactions (positive/negative) between them in normal tissue?

> This is a very good point from the reviewer. We also wondered if our data could result in a robust prediction of genetic interactions. Unfortunately, we could not define such prediction.

Do NUDIX enzymes with frequent epistatic interactions (identified in the normal cells) tend to change their RNA/protein expression level (compared to the normal tissue) more frequently in different cancer cell lines?

> We thank the reviewer for this comment. We have generated a new panel (Figure 5E) in which we compare mRNA expression with epistatic

interactions. The results show that NUDIX enzymes with strongly negative epistatic interactions identified in the normal cells (CCD841) tend to substantially decrease their mRNA expression levels in different cancer tissues, while NUDIX enzymes with strongly positive epistatic interactions in these normal cells tend to increase their mRNA expression levels across cancer tissues, although this effect is less pronounced

Based on their large-scale genetic interaction and gene-gene relationship data can the authors predict one or two potentially good target pairs from the NUDIX enzymes for intervention, e.g., treatment of the different cancer cell lines?

> Despite obtaining clear phenotypes in which the double depletion of two NUDIX hydrolases seem to be lethal for cancer cells in vitro, as well as showing genetic interactions, predicting potential targets for intervention is outside the scope of this manuscript. Instead, we would encourage the readers interested in the NUDIX hydrolases to further explore the many possibilities presented in our work.

Reviewer #5 (Remarks to the Author):

In the submitted manuscript, Puigvert et al. report an in-depth analysis of the human NUDIX genes, a family of hydrolases with mostly unknown biological function. The authors have integrated different datasets from own experiments and public databases to create a comprehensive NUDIX gene map, aiming at providing an insight into the biological functions of and possible redundancies between NUDIX enzymes.

The authors thoroughly describe their analyses and visualize their findings in very complex figures. They have collected an impressive dataset and use it to cluster the human NUDIX genes. The methods applied seem to be carefully chosen. However, the overall findings of the paper stay a little behind the expectations that the abstract raises. In general, text and figures are hard to read and to understand and it is difficult to extract the findings and judge the methods, especially if someone has a different research background.

Specific points:

1. The text is in parts hard to read and some sentences are confusing and need multiple readings in order to get the content. The language is sometimes sloppy and there are sentences with grammar mistakes or missing words. Some examples are found in lines 385, 529-536, see

also points 8 and 9. The manuscript should be carefully edited and checked for spelling mistakes and comprehensibility.

> We apologize for this issue; the text has been heavily revised accordingly.

2. The figures are too complex and some of them are very unintuitive (especially, but not limited to, Figures 2b, 3a, 3c, 6c, 8b). The manuscript would benefit from condensing the figures and trying to present important differences (e.g. Figures 3a, 3c). Showing all data available is well meant, but does not help and some information could be moved to the supplements.

> We are thankful for the reviewers' comment, however in order to provide an overview of the entire NUDIX family, we have decided to include all data, specifically in Figure 3. However, we have made an effort to improve the clarity of most of the figures.

3. In lines 314-316, it is claimed that the three cancer cell lines used share a substantial amount of genetic interactions, while the respective figure (5C) shows the exact opposite. Some overlap is always there and is to be expected by chance. Most of the interactions found are specific to the cell lines.

> We agree with the reviewer, applying a more stringent cut-off for the significance of the genetic interactions resulted in lower common genetic interactions, indicating that most of the interactions are cell specific.

4. The statistics behind the epistasis analyses described in results and methods remain somewhat unclear. It should be clearly stated how many replicates were made, what the significance cut-off of each test was (p values in the figures!) and at what false discovery rate the interactions were called.

> We have now clarified in the text the number of experimental replicates, as well as added a Supplemental Figure 4 to demonstrate the robustness of the knockdown experiments.

5. Along that line, it seems that in Figures 5A and 6D not only significant interactions are shown, but all deviations from the expected value are depicted in color code, giving the visual impression that almost all genes share interactions. As mentioned in the text, true interactions are rare and an interaction matrix is rather sparse, especially when considering such a small set of genes.

> We have now applied a more stringent cut-off for the significance of the interactions, which has been also clarified in the text. We assume the reviewer meant Figure 6A as there is no Figure 6D.

6. The cell cycle based genetic interactions shown for the cell cycle phenotypes show a clear difference between the non-cancer cell line (many/mostly increase in cell fraction) and the other cell lines (mostly decrease in cell fraction). This difference can also be an analysis artifact and is not properly discussed. Also, in cell line A549, NUDT5 suddenly shares an increase interaction with almost all others, but all decrease interactions are gone.

> We agree with the reviewer that the cell cycle genetic interactions largely differ between the non-cancer cells and the cancer cells. However, rather than being an artifact, this could be due to the slower cell cycle of the non-cancer cells, in which the effect of the NUDIX depletion is translated into milder changes in the cell cycle. We have addressed this point in the text.

7. Title of Figure 2 names information about cell cycle regulation, which is not found in the figure.

> We have corrected this issue.

8. Legend of Figure 3 names figures A-B, B, C instead of A-B, C, D.

> The nomenclature of the figure collides with the figure names. While the figure has panels A, B and C. Panel B contains A-B, C-D, E-F and so on, which refer to the tissue images.

9. Legend of Figure 5: It should be 'A genetic interaction was assigned to pairS of genes if the viability phenotype_ of the double knockdown...'

> Corrected

10. The paper lacks a proper comparison between all the different clusters of NUDIX genes that were found with different methods. The differences between structure and enzyme activity clusters are discussed, but in the end the reader is left to put together all the information. The information given in Figure 8 is very comprehensive, but in itself lacks hints to shared or different biological properties of the NUDIX genes/proteins. While condensing some of the figures, an additional summary of the found clusters and their differences to the clusters that came out of each dataset would be of benefit.

> We agree with the reviewer, given the large amount of data that we

generated, a summary would be very helpful to the reader. Therefore, we have added an extensive summary figure in which all results are collected for each NUDIX.

11. In order to really deduct function, it would be a good approach to see how the NUDIX genes cluster within a larger set of genes, not which clusters form in the family. Of course, a large-scale epistasis analysis is beyond the scope of this paper, but for most of the analyzed databases, information is available for huge numbers of genes representing different biological functions and processes. Clustering of NUDIX genes to different processes based on the comprehensive data analyzed could really point to biological function and redundancies.

> We are very thankful for the reviewer's comments and suggestion. In fact, we initially attempted to do such pathway and biological processes assessment. We would expect that this kind of analysis would result in a broad overview of the biological processes in which the NUDIX hydrolases are involved. Unfortunately, the amount of connectivity data from the NUDIX hydrolases to immediate connected genes is rather sparse, very limited and in most cases non-existing. Therefore, and unfortunately, we decided not to perform such analysis. We hope that as the amount of data regarding the interaction of NUDIX hydrolases with other proteins increases, such analysis will be in reach.

Reviewers' Comments:

Reviewer #2 (Remarks to the Author):

The revised manuscript is an improvement on the original and most of the points raised in my review have been addressed. The relative emphasis of the various findings is better and the discussion of the limitations of the study and the summary of the major conclusions regarding each nudix hydrolase make it more accessible to the reader. I still have some reservations about the concept of epistasis as proposed by the authors – see my previous comments about the possible artificial nature of this and the relevance only to knocked down cells – and also the lack of confirmation of the extent of protein knockdown by Western blotting – see previous comment about low levels of enzyme perhaps being sufficient to maintain function. Therefore, I should like to see a greater acknowledgement of these possibilities in a revised text (perhaps in the Methodology Remarks section). I feel that the usefulness of the data as a resource for further study overcomes these as major issues.

There are however still a few minor points that need to be addressed by the authors. With my apologies, some of these points could have been raised after the first submission, but the sheer volume of data to be considered meant that they were inevitably overlooked. However, they should be very straightforward to correct. Numbers refer to line number in revised merged file:

40-41 Nudix hydrolases are not involved directly in purine base metabolism – all substrates are phosphorylated. Also, given the variety of substrates used by many family members, it seems odd to highlight phospholipid metabolism, as only one species-restricted family member is currently known to be involved in this. A more widespread “function” could be used instead.

60-61 The MutT homolog MTH1 was not the “founding member”. MutT itself was.

67 change “to DNA” to “in DNA”

71-72 “non-nucleotide polyphosphates of capped mRNAs” should be “non-nucleoside polyphosphates and capped mRNAs” as in original. Also note “nucleoside” instead of “nucleotide”

124-5 I am concerned by the use of reference 36 on Arabidopsis NUDT7 is support of “previously reported alignments” of human NUDT7/8/16/19. Regrettably, the numbering system of plant nudix hydrolases does not follow the mammalian system so AtNUDT7 (better known now as AtNUDX7) is not the ortholog of human NUDT7. It is in fact closest to human NUDT6 and should not align closely with these four human enzymes. Please correct and check any other

uses of plant references for similar misunderstandings. Also, the sequence similarity of animal NUDT7 and NUDT8 was previously noted in ref. 35.

186-7 As indicated in my original review, Ap4G, p4A and p4G are not really novel substrates for NUDT2. Although they may not all have been used with human NUDT2, they have with NUDT2 orthologs in other species (called Ap4A hydrolase before NUDT terminology) (see also up fig 17 b summary).

208 It is unlikely that human and mouse NUDT7 are so different that this can be used as an explanation for the failure to find substrates for the human enzyme. This would be better left as “unexplained”.

285 The actual supplemental figure number should be given instead of “results summary tables figure”. Also, it is important to note that subcellular localizations in the Human Protein Atlas are not 100% reliable and sometimes do not match published data. It would be helpful to include in the section on Methodology Remarks a statement to the effect that the authors are reliant on database information being accurate, something that is out of their control. However, the HPA classifies NUDT2 as nucleoplasmic, which does agree with published work, yet the figure in Supp. Fig. 17 lists it as mitochondrial. Is this an error or can it be justified? It would be worthwhile checking the others.

304 I do not think duplicate determinations can provide “high” reproducibility. “Good” might be better.

307 There is nothing in ref 51 to indicate that reduced viability after NUDT2 knockdown is “expected”. NUDT2 status has previously been shown to affect proliferation rate, not viability. Even complete knockout yields cells that are perfectly viable (ref 41).

335 “nucleotide pool imbalance or damage to DNA” is a misinterpretation of my previous comment. Nucleotide pool imbalance can lead to mismatches during DNA replication which can in turn lead to strand breaks or other lesions (depending on how they are repaired) which in turn can slow the cell cycle. Better to say “nucleotide pool imbalance which can lead to replication-slowing DNA lesions” or something like that.

604 replace “in occasions” with “on occasion”

637 Ref 67 refers to a single enzyme so is not an appropriate reference for this general statement.

unsigned

Reviewer #3 (Remarks to the Author):

The authors of the manuscript have significantly revised several chapters of the manuscript, thereby improving the support for their claims. They have adequately revisited the phylogenetic analysis and the methodology used for determining the sequence/structure-based grouping of the studied NUDIX enzymes. The addition of Sup. Fig. 17 is very helpful as a comprehensive map of all the important findings. There are still minor issues that need to be addressed- mainly small comment that the authors need to consider adding, as well as some sentences to be rephrased.

1. P. 4, L. 111 – P. 5 L113: Indeed, there are significant differences between the domain-only MSA-based tree, relative to the full-length MSA-based tree, but the vast majority of these differences are located in the branching points closer to the root. All main NUDIX groups are the same in both trees (NUDT7- NUDT8-NUDT16- NUDT19, NUDT14- NUDT15, NUDT1- NUDT15 & NUDT17, NUDT12- NUDT13 and NUDT3-NUDT4-NUDT10-NUDT11). This means that largely, grouping is in agreement, but the similarity relationships closer to the trees' roots are ambiguous. A small comment can thus be added to the discussion (currently the “domain only” vs. “full sequence” groupings are not mentioned in the Discussion). Please also add a description of the how the trees were rooted.
2. P. 6, L. 153: a redundant mention of the AG McLennan reference (appears in P. 3, L. 76).
3. P. 7, L. 168-L. 176: consider moving to Methods.
4. P. 8, L. 203: “and substrates as those” – “and substrates than those”.
5. P. 8, L. 208-210: please write the %ID between the human and mouse NUDT7 orthologs. This will help the readers in understanding the differences observed.
6. Sup. Fig. 2D: unclear why two blue bars (both representing NUDT3?) appear in the graph.
7. P. 9, L. 225: reference to Sup. Fig 2A should be changed to Sup. Fig. 1A.
8. P. 11, L. 285: Change “Results summary tables figure” to Sup. Fig. 17
9. P. 12, L. 304 – please rephrase.
10. P. 13, L. 315: consider mentioning that NUDT13 had also the strongest effect on CC841 viability, when compared to all other NUDT depletion experiments.
11. P. 13, L. 333: please rephrase “required for disease”.
12. Figure 6: color legend should be added to panel C.
13. Are red results discussed?
14. Sup. Fig 16,L. 6: redundant closing parenthesis.
15. P. 23, L. 567-571: lengthy sentence. Consider breaking it.
16. P. 24,L. 595: “subtle nuance” stands in opposition to the complexity shown by the authors. Consider rephrasing.

17. P. 29, L. 717-732: consider shortening this paragraph. Re-introduction of genetic interactions can be more concise.
18. P. 30, L. 755: Comment: The reason for the over-representation of NUDIX proteins of similar sequence in parallel pathways (as opposed to NUDIX proteins that share the same pathway) may be analogous to the general trends of sequence/function similarities when paralogs and orthologs are compared. Paralogs are generally less conserved than orthologs. The negative epistasis suggests overlapping biochemical function (that may be explained by high sequence similarity). Positive genetic interaction (co-occurrence in the same pathway) may be less selected for overlapping specificity (such a comment, if chosen to be added by the authors, might complement “implying that there is redundancy among the NUDIX superfamily”).
19. P. 32, L. 789-805: the discussion here repeats many of the points already present in results, hence the authors should consider shortening it.
20. P. 37, L. 919: “correctness” -> validation.
21. P. 38, L. 965: redundant comma.
22. P. 38, L. 979: “, for statistics refer to” -> “. For statistics refer to”.
23. Sup. Table 1: Some table boundaries are missing.
24. P. 39, L. 1046: Freemont->Fremont.
25. P. 40, L. 1053: Freemont->Fremont.
26. Sup. Table 7: The 2nd page in this document is not clear (Ensembl IDs of NUDIX proteins?).
27. P. 41, L. 1116: “was seen”->”was considered”
28. P. 42: Consider moving the FUSION-based clustering method description to be right after line 1141 (and maybe also change the FUSION description to be after the description of Red, to follow the analysis flow in the Results chapter)
29. P. 42, L. 1176-1179: repeats sentences in the Results. Consider consolidating.
30. P. 42, L. 1212: “edges”->”lines”.
31. P. 44, L. 1229 and 1249 are the same.
32. The WORD document of the supplementary material still shows tracked changes.

REVIEWERS' COMMENTS:

Reviewer #2 (Remarks to the Author):

The revised manuscript is an improvement on the original and most of the points raised in my review have been addressed. The relative emphasis of the various findings is better and the discussion of the limitations of the study and the summary of the major conclusions regarding each nudix hydrolase make it more accessible to the reader. I still have some reservations about the concept of epistasis as proposed by the authors – see my previous comments about the possible artificial nature of this and the relevance only to knocked down cells – and also the lack of confirmation of the extent of protein knockdown by Western blotting – see previous comment about low levels of enzyme perhaps being sufficient to maintain function. Therefore, I should like to see a greater acknowledgement of these possibilities in a revised text (perhaps in the Methodology Remarks section). I feel that the usefulness of the data as a resource for further study overcomes these as major issues.

- We agree with this comment and we have added a short sentence in the Methodology Remarks to emphasize the importance of complete protein depletion when considering epistatic interactions.

There are however still a few minor points that need to be addressed by the authors. With my apologies, some of these points could have been raised after the first submission, but the sheer volume of data to be considered meant that they were inevitably overlooked. However, they should be very straightforward to correct. Numbers refer to line number in revised merged file:

40-41 Nudix hydrolases are not involved directly in purine base metabolism – all substrates are phosphorylated. Also, given the variety of substrates used by many family members, it seems odd to highlight phospholipid metabolism, as only one species-restricted family member is currently known to be involved in this. A more widespread “function” could be used instead.

- This has been corrected in the text

60-61 The MutT homolog MTH1 was not the “founding member”. MutT itself was.

- This has been corrected in the text

67 change “to DNA” to “in DNA”

- This has been corrected in the text

71-72 “non-nucleotide polyphosphates of capped mRNAs” should be “non-nucleoside polyphosphates and capped mRNAs” as in original. Also note “nucleoside” instead of “nucleotide”

- This has been corrected in the text

124-5 I am concerned by the use of reference 36 on Arabidopsis NUDT7 is support of “previously reported alignments” of human NUDT7/8/16/19. Regrettably, the numbering system of plant nudix hydrolases does not follow the mammalian system so AtNUDT7 (better known now as AtNUDX7) is not the ortholog of human NUDT7. It is in fact closest to human NUDT6 and should not align closely with these four human enzymes. Please correct and check any other uses of plant references for similar misunderstandings. Also, the sequence similarity of animal NUDT7 and NUDT8 was previously noted in ref. 35.

- Reference 36 is no longer used to support the alignment between human NUDT7, NUD8, NUDT16 and NUDT19. And we acknowledge the previous mammal NUDT7 and NUDT8 similarity.

186-7 As indicated in my original review, Ap4G, p4A and p4G are not really novel substrates for NUDT2. Although they may not all have been used with human NUDT2, they have with NUDT2 orthologs in other species (called Ap4A hydrolase before NUDT terminology) (see also up fig 17 b summary).

- This has been clarified in the text.

208 It is unlikely that human and mouse NUDT7 are so different that this can be used as an explanation for the failure to find substrates for the human enzyme. This would be better left as “unexplained”.

- For space constrains, this has been moved to the Methodology Remarks section and is no longer used to justify the lack of our NUDT7 activity.

285 The actual supplemental figure number should be given instead of “results summary tables figure”. Also, it is important to note that subcellular localizations in the Human Protein Atlas are not 100% reliable and sometimes do not match published data. It would be helpful to include in the section on Methodology Remarks a statement to the effect that the authors are reliant on database information being accurate, something that is out of their control. However, the HPA classifies NUDT2 as nucleoplasmic, which does agree with published work, yet the figure in Supp. Fig. 17 lists it as mitochondrial. Is this an error or can it be justified? It would be worthwhile checking the others.

- We agree with the reviewer’s comments, the data that we used for the subcellular localization summary relies on the quality and accuracy of the data available in the databases. We have noted this in the Methodology Remarks. Our classification of NUDT2 is based on the information available at UniProt, as well as own unpublished observations from our lab.

304 I do not think duplicate determinations can provide “high” reproducibility. “Good” might be better.

- “Good” has been used instead of “high” in the text.

307 There is nothing in ref 51 to indicate that reduced viability after NUDT2 knockdown is “expected”. NUDT2 status has previously been shown to affect proliferation rate, not viability. Even complete knockout yields cells that are perfectly viable (ref 41).

- We have replaced “viability” by “proliferation” to the sentence containing this reference.

335 “nucleotide pool imbalance or damage to DNA” is a misinterpretation of my previous comment. Nucleotide pool imbalance can lead to mismatches during DNA replication which can in turn lead to strand breaks or other lesions (depending on how they are repaired) which in turn can slow the cell cycle. Better to say “nucleotide pool imbalance which can lead to replication-slowing DNA lesions” or something like that.

- This has been clarified in the text.

604 replace “in occasions” with “on occasion”

- This has been corrected in the text.

637 Ref 67 refers to a single enzyme so is not an appropriate reference for this general statement.

- Two more references have been added to back up this statement.

unsigned

Reviewer #3 (Remarks to the Author):

The authors of the manuscript have significantly revised several chapters of the manuscript, thereby improving the support for their claims. They have adequately revisited the phylogenetic analysis and the methodology used for determining the sequence/structure-based grouping of the studied NUDIX enzymes. The addition of Sup. Fig. 17 is very helpful as a comprehensive map of all the important findings. There are still minor issues that need to be addressed- mainly small comment that the authors need to consider adding, as well as some sentences to be rephrased.

1. P. 4, L. 111 – P. 5 L113: Indeed, there are significant differences between the domain-only MSA-based tree, relative to the full-length MSA-based tree, but the vast majority of these differences are located in the branching points closer to the root. All main NUDIX groups are the same in both trees (NUDT7- NUDT8-NUDT16- NUDT19, NUDT14-NUDT15, NUDT1- NUDT15 & NUDT17, NUDT12- NUDT13 and NUDT3-NUDT4-NUDT10-NUDT11). This means that largely, grouping is in agreement, but the similarity relationships closer to the trees' roots are ambiguous. A small comment can thus be added to the discussion (currently the "domain only" vs. "full sequence" groupings are not mentioned in the Discussion). Please also add a description of the how the trees were rooted.

- A description of how the trees were rooted has been added to the materials and methods section, within the Multiple alignment and phylogenetic analysis (in short: the trees are unrooted).

- We agree with the reviewer, it is indeed correct that there are similarities in the major groupings of the full length and domain only trees. As the trees are both unrooted (a fact now noted in the manuscript), the location of the differences, or similarities have no evolutionary significance. The groupings in unrooted trees are useful in determining the conservancy and/or diversity of a set of sequences. Therefore, we believe that our statements concerning the difference between the trees and our reasoning for presenting the full length tree in the primary figure are sufficient.

2. P. 6, L. 153: a redundant mention of the AG McLennan reference (appears in P. 3, L. 76).

- The redundant references have been removed

3. P. 7, L. 168-L. 176: consider moving to Methods.

- We agree with the reviewer, this paragraph has been moved to the Materials and Methods section.

4. P. 8, L. 203: "and substrates as those" – "and substrates than those".

- The text has been corrected

5. P. 8, L. 208-210: please write the %ID between the human and mouse NUDT7 orthologs. This will help the readers in understanding the differences observed.

- For space constrains, this sentence has been moved to the Methodology Remarks section and is no longer used to justify the lack of our NUDT7 activity. Although we agree with the reviewers' suggestion.

6. Sup. Fig. 2D: unclear why two blue bars (both representing NUDT3?) appear in the graph.

- The colored bars correspond to the different NUDIX tested as indicated in the legend, left or right bars correspond to the concentration of these NUDIX, either at 5 or 200nM. At 5nM only NUDT3 gave signal (the others are so low that are almost non visible), while the rest did only give signal at 200nM. We hope this clarifies the confusion.

7. P. 9, L. 225: reference to Sup. Fig 2A should be changed to Sup. Fig. 1A.

- The correct reference should be Sup. Fig 1b, it is now correct.

8. P. 11, L. 285: Change "Results summary tables figure" to Sup. Fig. 17

- This has been corrected.

9. P. 12, L. 304 – please rephrase.

- The text has been rephrased

10. P. 13, L. 315: consider mentioning that NUDT13 had also the strongest effect on CC841 viability, when compared to all other NUDT depletion experiments.

- A phrase has been added to emphasize the difference NUDT13 and the rest of NUDIX depletion experiments in CCD841.

11. P. 13, L. 333: please rephrase "required for disease".

- The text has been rephrased accordingly

12. Figure 6: color legend should be added to panel C.

- We thank the reviewers' comment, but we think that a color legend would increase the complexity of the figure.

13. Are red results discussed?

- The red results are reported in the results section.

14. Sup. Fig 16,L. 6: redundant closing parenthesis.

- This has been fixed.

15. P. 23, L. 567-571: lengthy sentence. Consider breaking it.

- The text has been revised and shortened.

16. P. 24,L. 595: "subtle nuance" stands in opposition to the complexity shown by the authors. Consider rephrasing.

- While we appreciate the reviewer's input, we respectfully disagree with this critique. Something can be both complex and nuanced at the same time. While we have been able to discern the differences of the NUDIX family proteins and present those differences here these differences are in fact nuanced and not immediately obvious for a family with a conserved motif and partially conserved domain.

17. P. 29, L. 717-732: consider shortening this paragraph. Re-introduction of genetic interactions can be more concise.

- The paragraph has been re-phrased and shortened.

18. P. 30, L. 755: Comment: The reason for the over-representation of NUDIX proteins of similar sequence in parallel pathways (as opposed to NUDIX proteins that share the same pathway) may be analogous to the general trends of sequence/function similarities when paralogs and orthologs are compared. Paralogs are generally less conserved than orthologs. The negative epistasis suggests overlapping biochemical function (that may be explained by high sequence similarity). Positive genetic interaction (co-occurrence in the same pathway) may be less selected for overlapping specificity (such a comment, if chosen to be added by the authors, might complement "implying that there is redundancy among the NUDIX superfamily").

- We agree with the reviewer's comment and have added a complementary phrase

19. P. 32, L. 789-805: the discussion here repeats many of the points already present in results, hence the authors should consider shortening it.

- The discussion, as well as other parts of the manuscript, has been shortened accordingly.

20. P. 37, L. 919: "correctness" -> validation.

- This has been corrected in the text

21. P. 38. L. 965: redundant comma.

- This has been corrected in the text

22. P. 38. L. 979: ", for statistics refer to" -> ". For statistics refer to".

- This has been corrected in the text

23. Sup. Table 1: Some table boundaries are missing.

- This has been fixed

24. P. 39, L. 1046: Freemont->Fremont.

- This has been corrected in the text

25. P. 40, L. 1053: Freemont->Fremont.

- This has been corrected in the text

26. Sup. Table 7: The 2nd page in this document is not clear (Ensembl IDs of NUDIX proteins?).

- Indeed, this is correct, the NUDIX IDs are Ensembl Gene IDs, it has been clarified in the table.

27. P. 41, L. 1116: "was seen"->"was considered"

- This has been corrected in the text

28. P. 42: Consider moving the FUSION-based clustering method description to be right after line 1141 (and maybe also change the FUSION description to be after the description of Red, to follow the analysis flow in the Results chapter)

- The descriptions have been re-arranged to follow a better flow accordingly.

29. P. 42, L. 1176-1179: repeats sentences in the Results. Consider consolidating.

- The text has been revised accordingly

30. P. 42. L. 1212: "edges"->"lines".

- This has been corrected in the text

31. P. 44, L. 1229 and 1249 are the same.

- This has been corrected in the text

32. The WORD document of the supplementary material still shows tracked changes.

- This will be addressed when resubmitted.